# InfiGFusion: Graph-on-Logits Distillation via Efficient Gromov-Wasserstein for Model Fusion

**Yuanyi Wang**[1], **Zhaoyi Yan**[2], **Yiming Zhang**[1], **Qi Zhou**[1], **Yanggan Gu**[1],
**Fei Wu**[3], **Hongxia Yang**[1,2*]
[1]The Hong Kong Polytechnic University, [2]InfiX.ai, [3]Zhejiang University
yuan-yi.wang@connect.polyu.hk, hongxia.yang@polyu.edu.hk

## Abstract

Recent advances in large language models (LLMs) have intensified efforts to fuse heterogeneous open-source models into a unified system that inherits their complementary strengths. Existing logit-based fusion methods maintain inference efficiency but treat vocabulary dimensions independently, overlooking semantic dependencies encoded by cross-dimension interactions. These dependencies reflect how token types interact under a model's internal reasoning and are essential for aligning models with diverse generation behaviors. To explicitly model these dependencies, we propose **InfiGFusion**, the first structure-aware fusion framework with a novel *Graph-on-Logits Distillation* (GLD) loss. Specifically, we retain the top-$k$ logits per output and aggregate their outer products across sequence positions to form a global co-activation graph, where nodes represent vocabulary channels and edges quantify their joint activations. To ensure scalability and efficiency, we design a sorting-based closed-form approximation that reduces the original $O(n^4)$ cost of Gromov-Wasserstein distance to $O(n \log n)$, with provable approximation guarantees. Experiments across multiple fusion settings show that GLD consistently improves fusion quality and stability. InfiGFusion outperforms SOTA models and fusion baselines across 11 benchmarks spanning reasoning, coding, and mathematics. It shows particular strength in complex reasoning tasks, with +35.6 improvement on Multistep Arithmetic and +37.06 on Causal Judgement over SFT, demonstrating superior multi-step and relational inference.

## 1 Introduction

Recent advances in LLMs have sparked growing interest in aggregating diverse model capabilities to build stronger, more general-purpose systems. Existing collective approaches span a broad design space. Prompt-based techniques [1, 2] dynamically chain or compose pretrained LLMs with tools or APIs to expand functional scope. Multi-agent systems [3, 4] employ multiple LLMs to collaborate or debate, often outperforming single-agent models on complex reasoning tasks. Model ensembling [5, 6] aggregates outputs from different models to boost accuracy and robustness, while Mixture-of-Experts (MoE) [7, 8, 9] architectures improve scalability by activating specialized subnetworks. Meanwhile, parameter-level merging models [10, 11, 12] aim to consolidate multiple fine-tuned checkpoints into a single model. Each of these paradigms explores a different trade-off between performance, efficiency, and compatibility.

Among fusion strategies, model fusion via logit distillation [13, 14] has emerged as a flexible and efficient paradigm. It enables a single pivot model to absorb knowledge from multiple source models—often with diverse domains, sizes, or architectures—without increasing inference cost. A representative cross-tokenizer approach is ULD [15], which introduces a universal logit-distillation

---

[*]Corresponding author.

39th Conference on Neural Information Processing Systems (NeurIPS 2025).

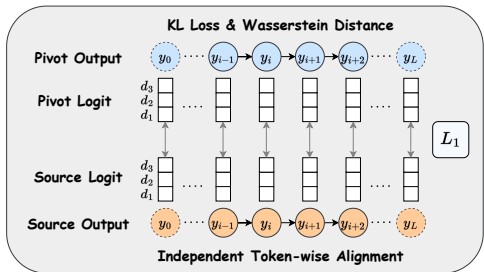 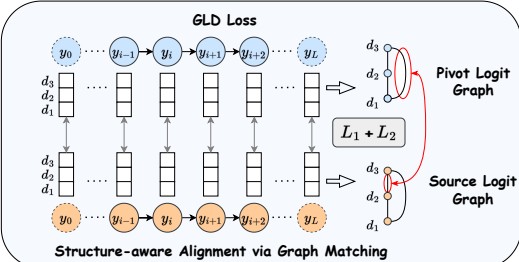

Figure 1: **Token-level vs. Structure-aware Fusion.** Given pivot and source logits of shape $[L, 3]$ (sequence length $L$, vocab size 3), token-level methods (left) align dimensions independently, ignoring token interactions. GLD (right) aggregates outer products into $[3, 3]$ co-activation graphs, capturing semantic dependencies via structure-aware graph alignment.

loss for aligning models with different tokenizers. Unlike traditional distillation, which transfers intermediate features or outputs, these methods align only the output logits using token-level objectives such as KL divergence or Wasserstein distance [16]. However, token-level methods inherently treat each vocabulary dimension independently, overlooking *semantic dependencies*—the co-activation patterns and relational structures among tokens that underlie a model's reasoning process. This limits their ability to align models with divergent generation behaviors, particularly for tasks requiring multi-step reasoning or fine-grained relational inference. Fig. 1 illustrates this limitation. While token-level losses enforce dimension similarity, they fail to capture the structured dependencies among tokens. InfiGFusion addresses this by modeling logits as graphs, aligning not just distributions but also the interactions that reflect a model's internal reasoning.

To address this challenge, we model semantic dependencies in the logit space using graph structures. Rather than treating token logits independently, we represent model outputs as feature-level graphs, where nodes correspond to token dimensions and edges capture their co-activation patterns across sequence positions. This representation reflects how a model internally organizes semantic relationships, beyond surface-level token probabilities, to support coherent reasoning and inference. Constructing such graphs from high-dimensional logits is nontrivial: vocabulary sizes are large, and most logits are near-zero [17, 18], making full pairwise modeling inefficient and noisy. To mitigate this, we apply top-$k$ sparsification to retain salient dimensions, followed by inner-product similarity to define edge weights. This yields interpretable graphs that faithfully reflect model reasoning behaviors.

Building on this, we propose **InfiGFusion**, a semantic structure-aware fusion framework that distills knowledge from multiple source models by aligning their logit graphs. At its core is the *Graph-on-Logits Distillation* (GLD) loss, which captures semantic dependencies by aligning structure-level information between source and pivot models. Specifically, we employ the Gromov-Wasserstein (GW) distance to match graph structures [19, 20]. However, standard GW incurs $\mathcal{O}(n^4)$ complexity, making it impractical for large vocabularies. To address this, we design a closed-form approximation based on node aggregation and sorting-based matching, reducing the cost to $\mathcal{O}(n \log n)$. We further provide theoretical guarantees on the approximation error, ensuring alignment fidelity.

We validate InfiGFusion across diverse fusion scenarios, demonstrating consistent improvements over state-of-the-art methods. By explicitly modeling semantic dependencies through logit graph alignment, InfiGFusion excels in tasks involving multi-step reasoning, relational inference, and fine-grained alignment—scenarios where token-level objectives fall short. Extensive experiments on 11 benchmarks, including reasoning, mathematics, and coding, confirm its effectiveness in producing more capable fusion models. Overall, our contributions can be summarized as follows:

- InfiGFusion is the first structure-aware fusion framework that models semantic dependencies via feature-level logit graphs and aligns them via a novel Graph-on-Logits Distillation loss.

- We propose a novel approximation to Gromov-Wasserstein distance, reducing complexity from $\mathcal{O}(n^4)$ to $\mathcal{O}(n \log n)$, with provable error bounds and rigorous theoretical analysis.

- Experiments on 11 benchmarks show that InfiGFusion consistently outperforms baselines, with significant gains on complex reasoning tasks (+35.6 on Multistep Arithmetic, +37.06 on Causal Judgement), demonstrating superior multi-step and relational inference.

## 2 Related Work

**Collective LLM:** To aggregate the capabilities of specialized LLMs, several strategies have been proposed. *Prompt-based integration* methods, such as Toolformer [1], ReAct [21], and HuggingGPT [22], guide LLMs to invoke tools or auxiliary models through natural language prompts. *Multi-agent collaboration* frameworks like ChatArena [23] and AgentBench [24] treat LLMs as interacting agents that debate or coordinate to improve reasoning. *Model ensembling* aggregates outputs via voting [25] or reranking, but suffers from high inference cost. *Mixture-of-Experts (MoE)* models [7, 8, 9] activate sparse expert subsets per query, offering scalability but introducing routing complexity. *Parameter merging* methods [10, 11, 12] combine weights from fine-tuned models, but require architectural alignment. While these strategies explore different trade-offs between performance, efficiency, and flexibility, they generally assume the underlying models remain fixed. In contrast, we focus on *knowledge fusion*, where a pivot model learns from multiple sources during training.

**Model Fusion via Knowledge Distillation:** Traditional knowledge distillation (KD) transfers knowledge from a large teacher to a smaller student via features, logits, or output distributions [26, 27], primarily for compression or acceleration. In contrast, model fusion via KD aims to integrate multiple source models—often with varying architectures, scales, and vocabularies—into a single target of similar size. Recent methods [13, 16, 14] fuse multiple LLMs by aligning token distributions using KL divergence or Wasserstein distance. These approaches typically require vocabulary alignment and assume semantic compatibility at the token level. However, they overlook *semantic dependencies* among logit dimensions—i.e., how token probabilities interact under internal reasoning, which are crucial for preserving logical structure. Our work addresses this gap by introducing a structure-aware distillation objective that explicitly aligns semantic dependencies via graph alignment. Unlike prior methods, it avoids explicit vocabulary alignment by modeling logits as graphs, enabling robust, vocabulary-agnostic fusion across heterogeneous models while preserving reasoning structures.

**Gromov-Wasserstein Distance:** The Gromov-Wasserstein (GW) distance [28, 20] is a relational optimal transport metric for aligning distributions based on internal structural similarity. It has been widely used in graph comparison [19, 29] and structure-aware representation learning [30, 31], especially when node correspondences are unknown. However, its standard formulation incurs $\mathcal{O}(n^4)$ complexity due to fourth-order interactions, rendering it impractical for large-scale logit graphs in LLMs. To mitigate this, prior works apply entropic regularization [20, 32], Sinkhorn-based solvers [19, 33], or low-rank projections [34]. While effective for small graphs, these methods lack closed-form solutions and suffer from convergence instability, making them less suited for distillation over vocabulary-sized graphs ($n > 10^5$). Our method takes a non-iterative, sorting-based approach: it compresses each similarity matrix into node-level features and aligns them via sorted-based matching, reducing complexity to $\mathcal{O}(n \log n)$. Unlike prior approximations, it yields a differentiable, training-friendly loss with a provable approximation bound. To our knowledge, this is the first closed-form, theoretically guaranteed GW approximation tailored for aligning logit graphs in LLM fusion.

## 3 Method

**InfiGFusion** fuses heterogeneous LLMs by distilling structured knowledge from multiple source models into a unified pivot model. Central to this process is the Graph-on-Logits Distillation (GLD) loss, which models logit vectors as feature-level graphs and aligns their structures across models.

To capture semantic dependencies, we adopt Gromov-Wasserstein distance for structure-level alignment between logit graphs. However, the standard GW incurs $\mathcal{O}(n^4)$ complexity, prohibitive for LLM-scale vocabularies ($n > 10^5$). We propose a sorting-based approximation that reduces this to $\mathcal{O}(n \log n)$ by compressing graph structures into node-level features. Furthermore, we provide a provable error bound of $\frac{n-1}{n^2} + \frac{m-1}{m^2}$ (Proposition 1), ensuring faithful and efficient graph alignment.

### 3.1 Overview of InfiGFusion

InfiGFusion integrates knowledge from diverse source models $\{M_1, \ldots, M_S\}$ into a unified pivot model $M_0$, enhancing reasoning capabilities without increasing inference cost.

Given an instruction-response dataset $\mathcal{D}$, $M_0$ is trained to align with both human supervision and source model behaviors via two complementary objectives: (i) *Universal Logit Distillation (ULD)*: a coarse token-level alignment combining supervised fine-tuning (SFT) and a sorting-based Wasserstein-

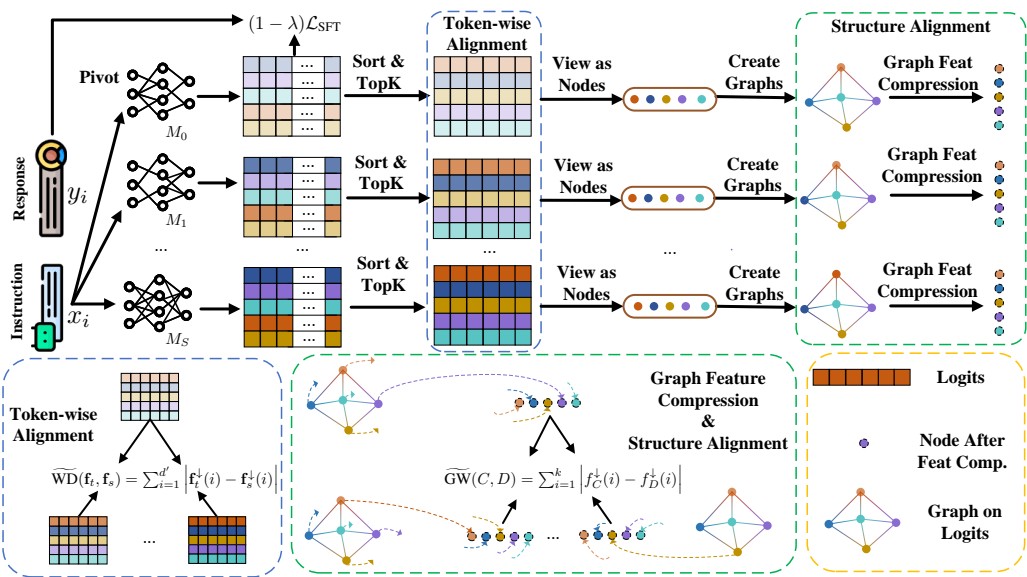

Figure 2: **InfiGFusion framework.** Given instruction-response pairs, source and pivot models produce logits, sparsified into feature-level graphs capturing semantic dependencies. We align graphs via an efficient Gromov-Wasserstein approximation (GLD), reducing complexity from $\mathcal{O}(n^4)$ to $\mathcal{O}(n \log n)$. The overall objective combines structure-aware distillation (GLD) with token-level distillation (ULD) and supervised signals (SFT) for robust fusion.

1 approximation [15]; (ii) *Graph-on-Logits Distillation (GLD)*: a structure-aware objective aligning relational dependencies in logit space. The overall loss is:

$$\mathcal{L}_{\text{InfiGFusion}} = \lambda_{\text{GLD}} \sum_{s=1}^{S} \mathcal{L}_{\text{GLD},s} + \lambda_{\text{ULD}} \sum_{s=1}^{S} \mathcal{L}_{\text{ULD},s} + \lambda_{\text{SFT}} \mathcal{L}_{\text{SFT}}, \tag{1}$$

where ULD ensures distributional alignment and GLD captures fine-grained reasoning structures. For token-level ULD, we adopt the linear-time Wasserstein-1 approximation [15], which compare the logit distributions between pivot model $t$ and source models $s$:

$$\mathcal{L}_{\text{ULD},s} = \widetilde{\text{WD}}(\mathbf{f}_t, \mathbf{f}_s) = \sum_{i=1}^{d'} \left| \mathbf{f}_t^{\downarrow}(i) - \mathbf{f}_s^{\downarrow}(i) \right|, \tag{2}$$

where $\mathbf{f}^{\downarrow}$ denotes sorting. This provides an efficient coarse alignment signal, complementing GLD's structural matching.

## 3.2 Dynamic Sparse Graph Construction

To capture structural dependencies among logit dimensions while avoiding the cost of full $d \times d$ modeling ($d$ is the vocabulary size for each model), we adopt a two-stage strategy: top-$k$ sparsification followed by dynamic graph construction.

**Top-$k$ Logit Sparsification.** Given a logit tensor $\mathbf{Z} \in \mathbb{R}^{B \times L \times d}$, we extract the top-$k$ logits for each position to obtain $\mathbf{Z}^{\text{top}} \in \mathbb{R}^{B \times L \times d'}$, with $d' \ll d$. We then unify the top-$k$ dimensions across tokens to obtain a consistent $d'$-dimensional representation per sample.

**Dynamic Graph Construction.** To model the interdependencies among logit dimensions, graph construction can generally follow two paradigms: (i) token-wise similarity across the sequence, or (ii) feature-wise similarity among logit dimensions. The former focuses on token interactions, while the latter captures intrinsic correlations between semantic dimensions. In this work, we adopt the feature-wise approach, as it empirically yields better alignment performance and more effectively preserves structural information. Specifically, for each sample $b$, we construct a graph $G_h = (V, E)$

over the reduced logits $Z_b \in \mathbb{R}^{L \times d'}$, where each node corresponds to a selected logit dimension. The adjacency matrix $C_b \in \mathbb{R}^{d' \times d'}$ is defined via dot-product similarity across the sequence:

$$C_b(i,j) = \sum_{t=1}^{L} z_t(i) \cdot z_t(j), \quad C_b = \mathbf{Z}_b^{\text{top}^\top} \cdot \mathbf{Z}_b^{\text{top}}. \tag{3}$$

This construction approximates cosine similarity after normalization and is applied to both pivot and source models at every training step, resulting in dynamic, data-dependent graphs that evolve with model predictions.

### 3.3 Efficient Approximation Algorithm

Directly applying *Gromov-Wasserstein (GW)* distance for structure-level alignment incurs $\mathcal{O}(n^4)$ complexity, which is infeasible for LLM-scale graphs. We propose an efficient approximation by summarizing graph structures into node-level features and aligning them via sorting-based matching, reducing complexity to $\mathcal{O}(n \log n)$. This method maintains alignment fidelity with a provable error bound of $\frac{n-1}{n^2} + \frac{m-1}{m^2}$ (Proposition 1). Full derivations are provided in Appendix A.

Given intra-graph similarity matrices $C_s, C_t \in \mathbb{R}^{d' \times d'}$ from source and pivot models, the squared GW distance is defined as:

$$\text{GW}(C, D) = \min_{\gamma \in \Pi(\mu_s, \mu_t)} \sum_{i,j,k,l} |C(i,j) - D(k,l)|^2 \gamma_{ik} \gamma_{jl}. \tag{4}$$

Despite its expressiveness, the GW objective requires $\mathcal{O}(n^2 m^2)$ complexity due to its fourth-order tensor form, making it impractical for large logits ($n, m$ are the vocabulary sizes). To address this, we derive a closed-form approximation by compressing the graph structure into scalar features.

**Graph Feature Compression.** Our key insight is that the structure of each similarity matrix $C \in \mathbb{R}^{n \times n}$ and $D \in \mathbb{R}^{m \times m}$ can be summarized using scalar node-level features, where $n, m$ are vocabulary size. Specifically, we compute degree-style features:

$$f_C(i) = \frac{1}{n} \sum_{j=1}^{n} C(i,j), \quad f_D(k) = \frac{1}{m} \sum_{l=1}^{m} D(k,l), \tag{5}$$

which captures the overall importance or connectivity of each logit dimension in the graph. This reduces the original $d' \times d'$ structure to two vectors in $\mathbb{R}^{d'}$.

**Separable Relaxation.** Substituting these features into the original objective yields a simplified, separable form:

$$\widetilde{\text{GW}}(C, D) = \sum_{i,k} |f_C(i) - f_D(k)|^2 \gamma_{ik}, \tag{6}$$

where symmetry is assumed across $\gamma_{ik}$ and $\gamma_{jl}$. This relaxation significantly lowers computational cost. As analyzed in Appendix A and the following proposition, the error is theoretically bounded:

**Proposition 1** (Approximation Error Bound). *Let $C \in \mathbb{R}^{n \times n}$ and $D \in \mathbb{R}^{m \times m}$ be the similarity matrices derived from logit self-inner products and row-normalized such that $\sum_j C_{ij} = 1$ and $\sum_l D_{kl} = 1$. Then the absolute error between the exact and approximated GW distances satisfies:*

$$|GW(C, D) - \widetilde{GW}(C, D)| \leq \frac{n-1}{n^2} + \frac{m-1}{m^2}. \tag{7}$$

The proof is provided in Appendix A. In practice, the size of $n, m$ is more than $10^5$. This bound guarantees the error decays as $\mathcal{O}(1/n)$, enabling scalable use in large-vocabulary settings.

**Sorting-Based Matching.** To avoid solving for the optimal transport plan $\gamma$ explicitly, we take a direct strategy like ULD [15] that adopts a deterministic transport strategy based on sorting. Let $f_C^\downarrow, f_D^\downarrow$ be the descending-sorted versions of $f_C$ and $f_D$. We construct a matching that aligns the $i$-th largest entry of $f_C$ to the $i$-th largest entry of $f_D$:

$$\widetilde{\text{GW}}(C, D) = \sum_{i=1}^{k} \left| f_C^\downarrow(i) - f_D^\downarrow(i) \right|, \quad k = \min(d'_s, d'_t). \tag{8}$$

This effectively reduces the original optimization to an $\mathcal{O}(n \log n)$ sorting-based assignment.

**Stability Analysis.** Beyond efficiency, our approximation improves training stability. We compare the Lipschitz constants of common alignment losses and the proof is provided in Appendix C:

**Proposition 2** (Lipschitz Constants Comparison). *We have the following relationship between the Lipschitz constants (L) for the Gromov-Wasserstein, Wasserstein, and KL losses:*

$$L_{GW} = \mathcal{O}\left(\frac{R^3}{D}\right) < L_{WD} = \mathcal{O}(\sqrt{D}) < L_{KL} = \mathcal{O}(e^{RD}), \tag{9}$$

*for vocabulary size D and logit range R, which means the GW-guided training models have tighter generalization bounds.*

The logit range $R$ is usually smaller than 120 for 14B LLMs. For example, with vocabulary size $D = 150,000$, $R$ is observed in the range of $[40, 120]$ [35]. This suggests GW-based alignment yields the lowest sensitivity to perturbations, contributing to improved generalization and stable training.

### 3.4 Unified GLD Loss

To support multi-source fusion, we extend the GLD loss into a unified formulation. For each source model $M_s$, we compute a structure-aware alignment loss with respect to the pivot model $M_0$ and aggregate all source contributions:

$$\mathcal{L}_{\text{GLD}} = \sum_{s=1}^{S} \widetilde{\text{GW}}(C_s, C_0). \tag{10}$$

where $C_s$ and $C_0$ denote the similarity matrices of the source and pivot graphs, respectively.

This unified objective enables efficient and scalable distillation from multiple source models, while preserving both semantic and relational structures across models.

## 4 Experiments

We evaluate InfiGFusion on large-scale LLM fusion, focusing on integrating models with diverse architectures and reasoning styles while maintaining inference efficiency. Our experiments verify that GLD enables faithful and effective fusion beyond token-level objectives. InfiGFusion outperforms state-of-the-art fusion methods and strong open-source baselines, with significant improvements in multi-step reasoning and relational inference. Additional results are provided in the Appendix.

### 4.1 Experimental Setup

**Datasets.** We construct a novel multi-task training dataset comprising 130k examples across general reasoning, mathematics, and coding. (i) The 52K general data are sourced from Infinity-Instruct [36]. (ii) Our mathematical dataset comprises

| Types | General | Math | Code |
|---|---|---|---|
| Dataset | Infinity-Instruct | NuminaMath-1.5 | KodCode-V1-SFT-R1 |
| Original Size | 1.4M | 1.4M | 268k |
| Filtered Size | 52K | 39K | 39K |

Table 1: Statistics of the novel datasets.

39k pairs of math questions and corresponding answers. The questions are sourced from the $NuminaMath\_1.5^2$ dataset, while the answers are distilled from the DeepSeek-R1-671B model by the AM team[3]. $NuminaMath\_1.5$ represents the second iteration of the widely acclaimed NuminaMath[37] dataset. It offers high-quality data for competition-level mathematical problems across various domains, including Algebra, Geometry, Combinatorics, Calculus, Inequalities, Logic and Puzzles, and Number Theory. (iii) In the code generation domain, we utilized the KodCode-V1-SFT-R1 dataset [38], which contains approximately 268k samples. Each sample was fed into our pivot model, which was prompted to generate 5 random responses per input. The generated outputs were then passed through a sandbox-based evaluation system. For each sample, if at least one of the five responses failed, the sample was flagged for further consideration. From these flagged samples, we filtered and selected 39k high-quality examples for further distillation.

---

[2]https://huggingface.co/datasets/AI-MO/NuminaMath-1.5
[3]https://huggingface.co/datasets/a-m-team/AM-DeepSeek-R1-Distilled-1.4M

| Models | Math | | | Coding | | General Reasoning | | | Instruct. | Text Reasoning | | Avg | Model Size | GPU Hour |
|---|---|---|---|---|---|---|---|---|---|---|---|---|---|---|
| | GSM8K | MATH | ThQA | MBPP | HEval | BBH | ARC | MMLU | IFEval | DROP | HS | | | |
| **Pivot Model** | | | | | | | | | | | | | | |
| Phi-4 | 87.41 | 80.04 | 51.12 | 70.8 | 83.54 | 68.84 | 93.9 | 85.62 | 77.34 | 88.67 | 87.62 | 79.54 | 14B | ∼1.0M |
| **Source Models** | | | | | | | | | | | | | | |
| Qwen2.5-Instruct | 91.13 | 78.16 | 47.25 | 81.70 | 83.54 | 77.59 | 92.20 | 80.22 | **85.01** | 85.56 | 88.28 | 80.97 | 14B | ∼1.8M |
| Mistral-Small | **92.42** | 69.84 | 48.50 | 68.80 | 84.15 | 81.59 | 91.86 | 81.69 | 82.25 | 86.52 | **91.84** | 79.95 | 24B | ∼1.6M |
| Qwen2.5-Coder | 89.16 | 74.18 | 38.88 | **85.40** | **90.90** | 75.40 | 89.49 | 75.08 | 74.70 | 84.34 | 79.83 | 77.94 | 14B | ∼1.8M |
| **SFT** | | | | | | | | | | | | | | |
| Pivot-SFT | 89.99 | **82.96** | 54.50 | 77.86 | 87.80 | 67.23 | 93.56 | **86.21** | 77.70 | 89.44 | 87.76 | 81.36 | 14B | 120 |
| **Model Fusion via Distillation** | | | | | | | | | | | | | | |
| MiniLogit | 89.80 | 79.78 | 46.36 | 78.49 | 85.44 | 82.68 | 92.19 | 84.58 | 79.36 | 88.56 | 88.25 | 81.43 | 14B | 220 |
| FuseLLM | 90.24 | 80.25 | 53.52 | 79.28 | 84.00 | 77.62 | 92.08 | 83.92 | 78.56 | 88.74 | 87.81 | 81.46 | 14B | 225 |
| FuseChat | 91.21 | 77.52 | 51.88 | 81.80 | 84.15 | 83.37 | 93.56 | 84.23 | 78.90 | 89.23 | 87.42 | 82.12 | 14B | 650 |
| InfiFusion | 90.07 | 80.94 | 54.62 | 79.63 | 84.72 | 80.94 | 94.24 | 85.81 | 76.02 | 89.27 | 87.91 | 82.20 | 14B | 160 |
| **InfiGFusion** | 90.45 | 81.92 | **55.38** | 85.2 | 86.00 | **85.62** | **94.58** | 85.24 | 80.22 | **89.62** | 88.22 | **83.85** | 14B | 195 |

Table 2: Main results on various benchmarks.

| Reasoning Models Models Size | Qwen2.5-Instruct 14B | Mistral 24B | DeepSeek-R1-Distill-Qwen 14B | Phi4 14B | Phi4 (SFT) 14B | InfiGFusion 14B |
|---|---|---|---|---|---|---|
| Abstract Algebra | 73 | 65 | 85 | 82 | 86 (+4.0↑) | **88** (+6.0↑) |
| Marketing | 90.17 | 92.31 | 89.32 | 92.74 | 92.74 (+0.0) | **95.3** (+2.56↑) |
| International Law | 81.82 | 81.82 | 82.64 | 91.74 | 90.91 (-0.83↓) | 90.91 (-0.83↓) |
| Moral Scenarios | 71.96 | 68.6 | 73.41 | 75.75 | 74.75 (-1.0↓) | 73.97 (-1.78↓) |
| Virology | 52.41 | 49.4 | **54.22** | 53.01 | 53.61 (+0.6↑) | **54.22** (+1.21↑) |
| Formal Logic | 68.25 | 66.67 | **91.27** | 77.78 | 78.57 (+0.79↑) | 74.6 (-3.18↓) |
| Security Studies | 77.96 | 78.78 | 78.37 | 77.14 | 79.59 (+2.45↑) | **81.22** (+4.08↑) |
| logical_fallacies | 85.28 | 84.66 | 85.28 | 86.5 | 87.73 (+1.23↑) | **87.73** (+1.23↑) |
| Ruin Names | 82.8 | 76.8 | 39.6 | **88.8** | 88.0 (-0.8↓) | 88.4 (-0.4↓) |
| Tracking 7 Objects | 80.4 | 96.4 | 80.8 | 94.4 | 90.0 (-4.4↓) | **96.8** (+2.4↑) |
| Tracking 5 Objects | 79.2 | **99.2** | 82.8 | 96.8 | 95.6 (-1.2↓) | 96.8 (+0.0) |
| Logical Deduction 3 | 97.6 | **98.8** | 83.2 | 98.4 | 96.4 (-2.0↓) | 97.6 (-0.8↓) |
| Logical Deduction 5 | 80.4 | 82 | 86 | 85.6 | 92.4 (+6.8↑) | **94** (+8.4↑) |
| Logical Deduction 7 | 68.8 | 62.8 | 83.2 | 88.4 | 88.8 (+0.4↑) | **89.2** (+0.8↑) |
| Colored Objects | 93.6 | 90 | 87.6 | **96.4** | 96.8 (+0.4↑) | **96.4** (+0.0) |
| Multistep Arithmetic | 96.4 | 93.2 | 81.6 | 64 | 62 (-2.0↓) | **99.6** (+35.6↑) |
| Dyck Languages | 35.6 | 37.2 | 9.6 | 11.2 | 13.2 (+2.0↑) | **40.0** (+28.8↑) |
| Causal Judgement | 43.85 | 68.98 | 45.35 | 32.99 | 35.83 (+2.84↑) | **70.05** (+37.06↑) |
| Avg. 18 Tasks | 75.53 | 77.37 | 73.29 | 77.43 | 77.94 (+0.51↑) | **84.16** (+6.73↑) |

Table 3: Performance on complex reasoning tasks. These datasets involve multi-step logic, causal inference, event tracking, and semantic analysis. InfiGFusion consistently improves over SFT, especially in complex reasoning. More results (84 tasks) are provided in Appendix F.

**Models and Baselines.** We fuse three source models: Qwen2.5-14B [39], Qwen2.5-Coder-14B [39], and Mistral-24B [40], into a Phi-4 [41] pivot model. Fusion baselines include MiniLogit [42], InfiFusion [16], FuseLLM [13], and FuseChat [43]. Full configurations are provided in Table 2.

**Evaluation.** We assess fusion performance on 11 benchmarks spanning *reasoning* (BBH [44], ARC-C [45], MMLU [46]), *mathematics* (GSM8K [47], Math [48], TheoremQA [49]), *coding* (MBPP [50], HumanEval [51]), *text reasoning* (DROP [52], HellaSwag [53]), and *instruction following* (IFE-val [54]). Additional training details and results are provided in Appendix D.

## 4.2 Main Results

Table 2 summarizes the performance of **InfiGFusion** across eleven benchmarks, compared with the pivot model, source models, and representative model fusion methods.

**Comparison with Pivot and Source Models.** InfiGFusion consistently outperforms both the pivot model and source models in overall performance. Compared to the pivot model, it achieves a significant improvement (+2.53 avg), particularly on complex reasoning benchmarks such as BBH, ARC, and MMLU, where semantic richness and multi-step inference are crucial. For instance, InfiGFusion surpasses the pivot by +16.7 on BBH, demonstrating its ability to aggregate diverse reasoning styles beyond single-source capabilities. Compared to source models, InfiGFusion effectively integrates complementary strengths. While individual source models excel in isolated tasks (e.g., Mistral-Small

on BBH, Qwen2.5-Coder on HEval), they struggle with generalization across domains. InfiGFusion surpasses these models with a more balanced performance, avoiding their trade-offs and consolidating knowledge into a unified pivot.

**Comparison with Model Fusion Baselines.** Against existing fusion methods (MiniLogit, FuseLLM, FuseChat), InfiGFusion exhibits superior fusion quality. Notably, while other distillation methods provide moderate improvements over the pivot, they often suffer from degraded performance in high-dependency tasks (e.g., logical reasoning, multi-condition alignment). InfiGFusion consistently delivers better results on structurally challenging tasks, indicating its advantage in preserving inter-token dependencies during fusion. Furthermore, compared to simpler fusion objectives, the structure-aware GLD loss of InfiGFusion yields better alignment without sacrificing robustness.

**Efficiency and Resource Usage.** In terms of computational cost, InfiGFusion maintains competitive efficiency. Despite introducing structure-aware alignment, its GPU hours remain modest (195 GPU hours), substantially lower than multi-step fusion pipelines like FuseChat (650 GPU hours). Model size remains identical to the pivot (14B), ensuring no additional inference overhead.

### 4.3 Complex Reasoning Performance

We evaluate InfiGFusion on 18 reasoning datasets spanning BBH and MMLU, covering multi-step logic, structural dependencies, and fine-grained semantic alignment, as shown in Table.3. These tasks are known to challenge token-level fusion methods due to their complex reasoning path. Comprehensive results can be found in Appendix F.

**Improved Reasoning over SFT.** Compared to Phi4-14B after supervised fine-tuning (SFT), InfiG-Fusion delivers consistent gains, boosting the average accuracy from 77.94% to 83.79%. Notable improvements are observed in tasks demanding complex relational reasoning: *Multistep Arithmetic* (+34.8%) and *Causal Judgement* (+34.39%). This demonstrates InfiGFusion's ability to enhance multi-hop inference and causal understanding by explicitly modeling semantic dependencies.

**Structural Alignment Benefits.** InfiGFusion excels in *Tracking Shuffled Objects* and *Logical Deduction* tasks, where aligning token relations is critical. For instance, it surpasses Phi4-14B SFT on *Logical Deduction Five Objects* (+1.6%) and *Seven Objects* (+0.4%), showcasing robust structure-level fusion beyond token alignment.

**Preserving Pivot Strength.** InfiGFusion maintains competitive performance in knowledge-centric tasks like *International Law* and *Ruin Names*, matching or slightly improving upon SFT baselines. This indicates that InfiGFusion not only integrates external model knowledge but also preserves the pivot model's original strengths. These results confirm that InfiGFusion's structure-aware fusion consistently enhances reasoning capabilities compared to token-level methods.

### 4.4 Ablation Study

**Effect of Source Model Diversity.** Table 4 evaluates the effect of fusing different source models. Individually, Qwen-Coder (Qc), Qwen-Instruct (Qi), and Mistral (M) each provide moderate gains, reflecting their domain-specific strengths. Notably, combinations such as Qi-M further amplify reasoning (+3.57), math (+3.06), and cod-

| Model | Reasoning | Math | Coding | Avg |
|---|---|---|---|---|
| Pivot: Phi4 | 81.43 | 72.86 | 77.17 | 79.54 |
| Qc | 84.39 (+2.96) | 74.18 (+1.32) | 86.1 (+8.63) | 83.20 (+3.66) |
| Qi | 86.03 (+4.60) | 74.08 (+1.22) | 84.89 (+7.72) | 83.52 (+3.98) |
| M | 84.50 (+3.07) | 75.77 (+2.91) | 85.8 (+8.63) | 83.62 (+4.08) |
| Qc-M | 84.89 (+3.46) | 74.18 (+1.32) | 86.09 (+8.92) | 83.54 (+4.00) |
| Qi-M | 85.00 (+3.57) | 75.92 (+3.06) | 85.39 (+8.22) | 83.72 (+4.18) |
| InfiGFusion | 87.06 (+5.63) | 74.74 (+1.88) | 85.6 (+8.43) | 83.85 (+2.53) |

Table 4: Ablation study on model diversity. Qc, Qi, and M mean Qwen-Coder, Qwen-Instruct, and Mistral.

ing (+8.22) performance, demonstrating that complementary inductive biases from different models enhance the pivot model. InfiGFusion achieves the best results (+2.53 avg), confirming that our framework effectively consolidates diverse reasoning behaviors into a unified model.

**Effect of Loss Components.** Table 5 analyzes our loss design. Removing ULD (w/o ULD) leads to a significant drop (-1.52 avg), particularly in reasoning (-1.8), indicating that coarse-grained token-level alignment is crucial for providing a reliable starting point. ULD narrows large dis-

| Component | Reasoning | Math | Coding | Avg |
|---|---|---|---|---|
| GLD + ULD | 87.06 | 74.74 | 85.6 | 83.85 |
| w/o ULD | 85.26 (-1.8) | 73.28 (-1.46) | 85.28 (-0.32) | 82.33 (-1.52) |
| w/o GLD | 85.70 (-1.36) | 75.21 (+0.47) | 85.18 (-0.42) | 83.16 (-0.69) |

Table 5: Ablation study on loss components.

tribution mismatches, effectively preparing the model for the finer-grained structural alignment performed by GLD. Excluding GLD (w/o GLD), on the other hand, yields a smaller overall decline (-0.69 avg), but reasoning performance still drops notably (-1.36). This highlights that while ULD han-

dles first-order token matching, it lacks the capacity to capture higher-order semantic dependencies. When combined, ULD and GLD form a synergistic alignment mechanism: ULD aligns distributions coarsely and stabilizes optimization, while GLD refines alignment by enforcing consistency in the relational structure of logits. The empirical results validate this design, with the full loss achieving the best performance. All ablation studies are tested in the Top-$k = 10$ setting.

## 4.5 Effect of Top-$k$

InfiGFusion sparsifies logits by retaining top-$k$ token dimensions before graph construction, selecting the most salient indices per sequence position. This inductive bias suppresses noisy activations and emphasizes meaningful token dependencies, serving as the foundation for graph-based semantic alignment. We evaluate Top-$k \in \{5, 10, 15, 20, 25, 30\}$ and report the results in Fig. 3. Increasing $k$ from 5 to 10 brings notable gains across all tasks (Avg +0.54), as larger $k$ captures richer token interactions essential for semantic graph construction. However, further increasing $k$ beyond 10 yields diminishing returns and even slight degradations, particularly in reasoning (-0.49 from Top10 to Top30). This suggests that excessive low-confidence tokens introduce spurious edges, diluting graph discriminability. The observed trend aligns with the heavy-tailed nature of LLM logits [17],

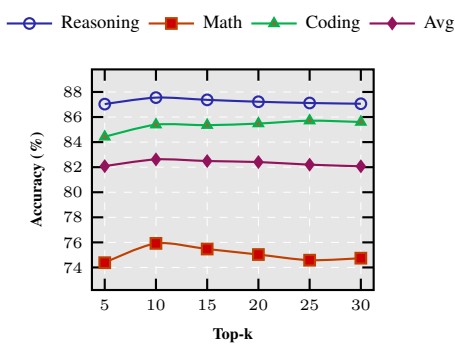

Figure 3: Top-k analysis.

where only a small subset of token dimensions are informative. Overall, Top-10 achieves the best trade-off, balancing informative graph structure and computational efficiency.

## 4.6 Case study

To illustrate InfiGFusion's strengths beyond token-level alignment, we analyze representative cases, comparing outputs from a standard SFT-tuned model (Phi4) and our InfiGFusion model. The selected examples highlight InfiGFusion's superior reasoning in multi-step causality and relational inference.

**Case 1: Frank T. Shooting Incident.** While both models predict "No," InfiGFusion performs a deeper step-by-step causality analysis. It explicitly identifies the causal chain's disruption—distinguishing between "intent," "misfire," and "accidental result"—showcasing robust *multi-step causality disambiguation* capabilities.

**Case 2: Wallace's Dual Cause of Death.** Unlike Phi4's surface-level judgment, InfiGFusion reasons through the latent causal structure, identifying "organized crime" as a distal cause. This reflects its strength in *relational and indirect causality inference*, where it effectively models complex event chains and upstream dependencies.

These cases demonstrate that InfiGFusion excels at: (i) *Multi-step inference:* Decomposing complex causal chains into interpretable reasoning steps. (ii)

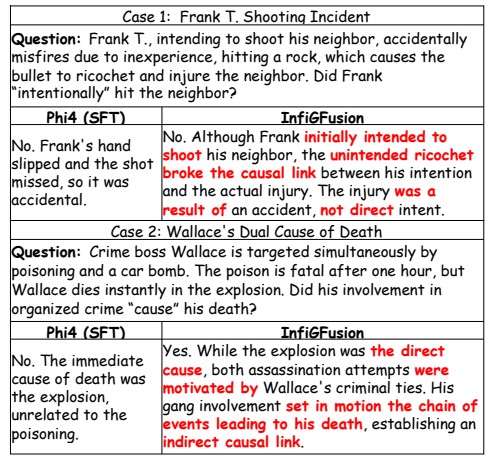

Figure 4: Case study.

*Relational causality reasoning:* Capturing indirect and upstream causal factors often missed by SFT models. (iii) *Fine-grained disambiguation:* Distinguishing intent, action, and outcome with structured alignment of reasoning behaviors.

## 4.7 Analysis of the GW approximation

In addition to the main results, we also conducted some experiments to analyze our proposed Gromov-Wasserstein approximation. It compares the quality of GW estimation in a simple simulated scenario. For example, by obtaining samples from two Gaussian distributions, and comparing the value of exact GW, sinkhorn-based approximation, and our proposed approximation. While our experiments

| $n$ | Exact&&Sinkhorn GW | Approx GW | Approx RE ($\pm$std) | Time Exact (s) | Time Sink (s) | Time Approx (s) |
|---|---|---|---|---|---|---|
| 50 | $0.4815 \pm 0.0452$ | $0.1978 \pm 0.0207$ | $0.5888 \pm 0.0242$ | $0.0171 \pm 0.0207$ | $0.0027 \pm 0.0005$ | $0.0004 \pm 0.0004$ |
| 100 | $0.3247 \pm 0.0297$ | $0.1261 \pm 0.0217$ | $0.6048 \pm 0.0907$ | $0.0072 \pm 0.0008$ | $0.0070 \pm 0.0005$ | $0.0001 \pm 0.0000$ |
| 200 | $0.2033 \pm 0.0088$ | $0.0973 \pm 0.0125$ | $0.5230 \pm 0.0426$ | $0.1390 \pm 0.0377$ | $0.0983 \pm 0.0033$ | $0.0359 \pm 0.0305$ |
| 500 | $0.1131 \pm 0.0050$ | $0.0587 \pm 0.0143$ | $0.4850 \pm 0.1121$ | $0.4405 \pm 0.1337$ | $0.4815 \pm 0.0904$ | $0.0607 \pm 0.0353$ |
| 1000 | $0.0764 \pm 0.0011$ | $0.0387 \pm 0.0133$ | $0.4921 \pm 0.1785$ | $2.3631 \pm 0.4981$ | $2.2456 \pm 0.4085$ | $0.1238 \pm 0.0399$ |
| 1500 | $0.0621 \pm 0.0028$ | $0.0308 \pm 0.0016$ | $0.5031 \pm 0.0398$ | $12.7151 \pm 4.1182$ | $9.1998 \pm 2.1952$ | $0.0982 \pm 0.0015$ |
| 2000 | $0.0496 \pm 0.0028$ | $0.0294 \pm 0.0058$ | $0.3995 \pm 0.1453$ | $20.1236 \pm 5.0250$ | $19.8941 \pm 8.1827$ | $0.0411 \pm 0.0424$ |
| 2500 | $0.0455 \pm 0.0008$ | $0.0305 \pm 0.0075$ | $0.3285 \pm 0.1712$ | $20.9198 \pm 5.6782$ | $30.1367 \pm 3.1038$ | $0.0026 \pm 0.0014$ |
| 3000 | $0.0404 \pm 0.0002$ | $0.0324 \pm 0.0141$ | $0.3941 \pm 0.0767$ | $52.8898 \pm 9.2735$ | $53.5049 \pm 12.4434$ | $0.2408 \pm 0.2562$ |

Table 6: GW estimation on synthetic Gaussians. In our runs, *Exact GW* and *Sinkhorn GW* are numerically indistinguishable, so we report them in a single column labeled *Exact&&Sinkhorn GW*.

demonstrate that the approximation is suitable as a training objective, it would be interesting to also understand its behavior as GW estimator.

**Experiment setup:** For each $n \in \{50, 100, 200, 500, 1000, 1500, 2000, 2500, 3000\}$, we sample $n$ points from $\mathcal{N}(0, I)$ and $\mathcal{N}(1, I)$, build their $n \times n$ Euclidean distance matrices, and compute: (i) *Exact GW* and *Sinkhorn GW* via `ot.gromov.gromov_wasserstein2(...,` `loss='square_loss')`; (ii) *Approx GW* via our sorting-based closed-form. Each configuration is repeated $3\times$ to report mean$\pm$std. *Approx RE* quantifies the approximation accuracy relative to the exact GW: $\mathrm{RE} = \frac{|\widehat{GW}_{\mathrm{approx}} - GW_{\mathrm{exact}}|}{GW_{\mathrm{exact}}}$ (mean$\pm$std over 5 seeds). In this symmetric Gaussian setting, Exact and Sinkhorn GW coincide to our reported precision (even with $\varepsilon = 10^{-1}$), evidencing the high accuracy of the Sinkhorn estimator here.

**Results and analysis:** Each setting is repeated 5 times with different seeds; we report mean$\pm$std. Across all sizes, *Sinkhorn GW* matches *Exact GW* to our reported precision (RE$\approx 0$), hence we merge them into a single column in Table 6. Our *Approx GW* exhibits decreasing relative error as $n$ grows (from $\approx 0.59$ at $n = 50$ down to $\approx 0.39$ at $n = 3000$), consistent with the $O(1/n)$ behavior. In terms of runtime, *Exact/Sinkhorn* scale poorly and rapidly become impractical beyond a few hundred points (already seconds at $n = 1000$ and tens of seconds by $n \geq 1500$ on our machine), whereas *Approx GW* remains below $\sim 0.3$ s even at $n = 3000$, demonstrating its $O(n \log n)$ scalability.

## 5 Conclusion

In this work, we propose InfiGFusion, a structure-aware fusion framework that explicitly models semantic dependencies among logits through a novel Graph-on-Logits Distillation (GLD) loss. By leveraging logit graph representations and an efficient Gromov-Wasserstein approximation, InfiGFusion enables the integration of heterogeneous LLMs without increasing inference cost. Extensive experiments on 11 benchmarks demonstrate that InfiGFusion consistently outperforms state-of-the-art fusion methods and baselines. In particular, it achieves notable gains in multi-step and relational reasoning tasks, such as +35.6 on Multistep Arithmetic and +37.06 on Causal Judgement over SFT models. It highlights the importance of structure-preserving alignment for effective model fusion in complex reasoning tasks, which is a promising direction for advancing collective LLM.

## Limitations

While InfiGFusion demonstrates clear advantages in tasks involving multi-step reasoning, causal inference, and relational alignment, its performance shows limitations in scenarios dominated by literal token matching or factual recall. For tasks where the correct answer is encoded by a few dominant logit dimensions (e.g., factoid QA), the added structural alignment of GLD brings marginal benefits and may even introduce noise from irrelevant dependencies. Moreover, when source models present conflicting factual knowledge, InfiGFusion focuses on preserving relational consistency but lacks explicit mechanisms for factual conflict resolution, sometimes amplifying semantic ambiguities. These limitations reflect inherent trade-offs in structure-aware fusion and motivate future work on adaptive dependency modeling and source reliability estimation.

Our model is available at https://huggingface.co/InfiX-ai/InfiGFusion-14B.

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

# A Appendix - Proof of the Approximation

**Step 1: Gromov-Wasserstein Distance Definition**

Consider two graphs with pairwise distance matrices within the respective graphs

$$C \in \mathbb{R}^{n \times n} \quad \text{and} \quad D \in \mathbb{R}^{m \times m},$$

and associated node distributions $a \in \Delta^n$ and $b \in \Delta^m$, where $\Delta^n$ denotes the probability simplex in $\mathbb{R}^n$. The squared Gromov–Wasserstein (GW) distance is defined as

$$\text{GW}(C, D, a, b)^2 = \min_{\gamma \in \Pi(a,b)} \sum_{i,j,k,l} |C_{ij} - D_{kl}|^2 \, \gamma_{ik} \, \gamma_{jl} \,,$$

with the transport plan $\gamma \in \mathbb{R}_+^{n \times m}$ satisfying the marginal constraints

$$\gamma 1_m = a \quad \text{and} \quad \gamma^\top 1_n = b \,.$$

Because of the inherent fourth-order interactions, computing the GW distance exactly is prohibitively expensive. Our goal is to derive a closed-form approximation by *compressing the high-dimensional structural information into node-level features* and then *aligning these features via a sorting-based matching strategy*.

**Step 2: Dimensionality Reduction**

The key observation is that the full distance matrices $C$ and $D$ encapsulate the network structure, but a significant part of this information may be captured by summarizing each node's relationships in **a single scalar** or **low-dimensional vector**. For example, one may define the node features by node degrees as:

$$f_C(i) = \frac{1}{n} \sum_{j=1}^{n} C_{ij} \quad \text{and} \quad f_D(k) = \frac{1}{m} \sum_{l=1}^{m} D_{kl} \,.$$

Alternatively, one might use the row norm or features derived from a diffusion kernel. In any case, the function $f$ serves to reduce the high-dimensional distance information into a quantity that reflects the "importance" or overall structure of each node. In our work, we apply the node features defined by node degrees.

**Step 3: Approximation of the Gromov-Wasserstein Distance**

Once the node-level features $f_C$ and $f_D$ are extracted, we want to approximate the GW formulation as follows:

$$\widetilde{\text{GW}}(C, D) = \sum_{i,j,k,l} |f_C(i) - f_D(k)| \cdot |f_C(j) - f_D(l)| \gamma_{ik} \, \gamma_{jl} \,.$$

It is obtained by approximating the element-wise cost in GW formulation and constraining the gap between the original and approximated formulation by a negligible constant.

**Proposition 1** [Approximation of the GW] Let $C \in \mathbb{R}^{n \times n}$ and $D \in \mathbb{R}^{m \times m}$ be the similarity matrices derived from logit self-inner products and row-normalized such that $\sum_j C_{ij} = 1$ and $\sum_l D_{kl} = 1$. Then the absolute error between the exact and approximated GW distances satisfies:

$$\left| \text{GW}(C, D) - \widetilde{\text{GW}}(C, D) \right| \leq \frac{n-1}{n^2} + \frac{m-1}{m^2} \tag{11}$$

*Proof.* Let $C \in \mathbb{R}^{n \times n}, \quad D \in \mathbb{R}^{m \times m}$ be two similarity matrices obtained by computing each model's logits' self-inner products and row-normalizing so that

$$\sum_{j=1}^{n} C_{ij} = 1, \quad C_{ij} \geq 0, \quad \sum_{l=1}^{m} D_{kl} = 1, \quad D_{kl} \geq 0.$$

Define their row–means and global means:

$$f_C(i) = \frac{1}{n} \sum_{j=1}^{n} C_{ij}, \quad \mu_C = \frac{1}{n^2} \sum_{i,j} C_{ij},$$

$$f_D(k) = \frac{1}{m} \sum_{l=1}^{m} D_{kl}, \quad \mu_D = \frac{1}{m^2} \sum_{k,l} D_{kl}.$$

Set
$$A = \sum_{i,j,k,l} \left| C_{ij} - D_{kl} \right|^2, \quad B = \sum_{i,j,k,l} \big(f_C(i) - f_D(k)\big)\big(f_C(j) - f_D(l)\big).$$

We expand each in turn:

**Expansion of $A$.**
$$A = \sum_{i,j,k,l} \left( C_{ij}^2 + D_{kl}^2 - 2\, C_{ij} D_{kl} \right)$$
$$= m^2 \sum_{i,j} C_{ij}^2 + n^2 \sum_{k,l} D_{kl}^2 - 2 \sum_{i,j,k,l} C_{ij} D_{kl}$$
$$= m^2 \sum_{i,j} C_{ij}^2 + n^2 \sum_{k,l} D_{kl}^2 - 2\, n^2 m^2 \, \mu_C \, \mu_D.$$

**Expansion of $B$.**
$$B = \sum_{i,j,k,l} \big(f_C(i) - f_D(k)\big)\big(f_C(j) - f_D(l)\big)$$
$$= \Big(\sum_{i,k}(f_C(i) - f_D(k))\Big) \Big(\sum_{j,l}(f_C(j) - f_D(l))\Big)$$
$$= \Big(m \sum_{i} f_C(i) - n \sum_{k} f_D(k)\Big)^2 = (nm\,(\mu_C - \mu_D))^2 = n^2 m^2\,(\mu_C - \mu_D)^2.$$

**Combine $A - B$.**
$$A - B = \Big[m^2 \sum C^2 + n^2 \sum D^2 - 2n^2 m^2 \mu_C \mu_D\Big] - n^2 m^2 (\mu_C - \mu_D)^2$$
$$= m^2 \sum C^2 + n^2 \sum D^2 - 2n^2 m^2 \mu_C \mu_D - n^2 m^2 (\mu_C^2 - 2\mu_C\mu_D + \mu_D^2)$$
$$= m^2 \sum C^2 + n^2 \sum D^2 - n^2 m^2 (\mu_C^2 + \mu_D^2 + 2\mu_C\mu_D) + 2n^2 m^2 \mu_C\mu_D - 2n^2 m^2 \mu_C\mu_D$$
$$= m^2 \sum C^2 + n^2 \sum D^2 - n^2 m^2 (\mu_C^2 + \mu_D^2).$$

Recall the global Root Mean Square (RMS) variance definitions
$$\Sigma_C^2 = \frac{1}{n^2} \sum_{i,j}(C_{ij} - \mu_C)^2, \quad \Sigma_D^2 = \frac{1}{m^2} \sum_{k,l}(D_{kl} - \mu_D)^2.$$

Hence
$$\sum_{i,j} C_{ij}^2 = n^2\big(\mu_C^2 + \Sigma_C^2\big), \quad \sum_{k,l} D_{kl}^2 = m^2\big(\mu_D^2 + \Sigma_D^2\big).$$

Substitute into $A - B$:
$$A - B = m^2\big(n^2(\mu_C^2 + \Sigma_C^2)\big) + n^2\big(m^2(\mu_D^2 + \Sigma_D^2)\big) - n^2 m^2(\mu_C^2 + \mu_D^2)$$
$$= n^2 m^2\, \Sigma_C^2 + n^2 m^2\, \Sigma_D^2 = n^2 m^2\big(\Sigma_C^2 + \Sigma_D^2\big).$$

Under uniform transport $\gamma_{ik} = 1/(nm)$,
$$\text{error} = \left| \sum_{i,j,k,l} (|C_{ij} - D_{kl}|^2 - \cdots)\, \gamma_{ik}\gamma_{jl} \right| = \frac{A - B}{n^2 m^2} = \Sigma_C^2 + \Sigma_D^2.$$

Since each row of $C$ sums to 1,
$$\sum_{j=1}^{n} C_{ij} = 1,$$

the worst–case per-row variance occurs when one entry is 1 and the rest 0:

$$\sum_{j=1}^{n}\left(C_{ij} - \tfrac{1}{n}\right)^2 = \left(1 - \tfrac{1}{n}\right)^2 + (n-1)\left(0 - \tfrac{1}{n}\right)^2 = \frac{n-1}{n}.$$

Summing over $i = 1, \ldots, n$ gives $\sum_{i,j}(C_{ij} - \tfrac{1}{n})^2 = n - 1$. Thus

$$\Sigma_C^2 = \frac{n-1}{n^2}, \quad \Sigma_D^2 = \frac{m-1}{m^2}.$$

Combining the above,

$$\left|\mathrm{GW}(C,D) - \widetilde{\mathrm{GW}}(C,D)\right| \;\leq\; \Sigma_C^2 + \Sigma_D^2 \;\leq\; \frac{n-1}{n^2} + \frac{m-1}{m^2}$$

$\square$

Because the summation is separable in the pairs $(i,k)$ and $(j,l)$, define

$$\Delta(\gamma) = \sum_{i,k}\left|f_C(i) - f_D(k)\right|\gamma_{ik}.$$

Thus, the approximate squared GW distance reduces to

$$\widetilde{\mathrm{GW}}^2 = (\Delta(\gamma))^2.$$

Our focus now turns to choosing a transport plan $\gamma$ that minimizes this expression. Notably, we only provide the proof of the scalar method (degree property as the feature), since the spectral method can be proved in the same way.

**Step 4: Choosing the Transport Plan via Sorting-Based Matching**

The transport plan $\gamma$ must satisfy the marginal conditions:

$$\sum_k \gamma_{ik} = a(i), \quad \sum_i \gamma_{ik} = b(k).$$

For simplicity, assume uniform marginals, i.e., $a(i) = \tfrac{1}{n}$ and $b(k) = \tfrac{1}{m}$. In this setting, $\gamma$ is a doubly stochastic matrix with a balanced assignment.

To bypass the computational difficulty of optimizing over all such matrices, we adopt a direct matching approach by sorting the node features. Let

$$f_C^{\downarrow} = \mathrm{sort}(f_C) \quad \text{and} \quad f_D^{\downarrow} = \mathrm{sort}(f_D),$$

with the sort performed in descending order. A natural matching then pairs the $i$−th largest entry of $f_C$ with the $i$−th largest entry of $f_D$.

This sorted matching corresponds to an induced transport plan that effectively acts as an indicator matrix. Under this matching, the transport cost simplifies to

$$\Delta(\gamma) \approx \sum_{i=1}^{k}\left|f_C^{\downarrow}(i) - f_D^{\downarrow}(i)\right|,$$

where $k = \min\{n, m\}$.

**Step 5: Final Solving Strategy**

By combining the above steps, we obtain the following approximation strategy for solving the GW distance:

$$\mathrm{GW}(C,D) \approx \sum_{i=1}^{k}\left|f_C^{\downarrow}(i) - f_D^{\downarrow}(i)\right|, \quad k = \min\{n, m\}.$$

Here, $f_C^{\downarrow}$ and $f_D^{\downarrow}$ denote the sorted node feature vectors (obtained in Step 2, like the row mean or diffusion kernel methods).

### A.1 Discussion

- **Dimensionality Reduction:** By condensing the full distance matrices to one-dimensional node features, the original fourth-order optimization problem is transformed into a much simpler vector matching problem.

- **Direct Matching:** The sorting-based strategy provides a natural one-to-one correspondence between nodes under the assumption of uniform marginals, thus eliminating the need for expensive optimization over the transport plan space.

- **Closed-Form Expression:** The final formulation is computationally efficient and differentiable, making it suitable for token space alignment.

- **Flexibility:** Although we illustrated the process with a simple row-mean (equal to the degree property) extractor, alternative feature extractors (e.g., row norm or diffusion-based features) can be used to potentially improve the approximation quality.

## B  Appendix - Error Analysis of the Closed-Form

This section aims to investigate whether the approximate GW distance converges to the true GW distance when the node features become increasingly representative of the underlying graph structure.

### B.1  Part 1: Stability of Feature Extraction

We want to show the *degree feature extraction* (i.e. taking row means of the similarity matrix) in our work is **Lipschitz continuous**, and hence, under mild conditions, the extracted features will converge to the true underlying structural descriptors as the noise decreases or with increasing sample size. We take the row means (degree property) strategy as an example and give the specific analysis, where the spectral strategy can be analyzed in the same way.

#### B.1.1  Notations

Let $C \in \mathbb{R}^{n \times n}$ be the similarity matrix, representing graph structure, and define the degree feature of node $i$ by

$$f(i) = \frac{1}{n} \sum_{j=1}^{n} C_{ij} \,.$$

This defines the mapping

$$F : \mathbb{R}^{n \times n} \to \mathbb{R}^n, \quad F(C) = \frac{1}{n} C \mathbf{1} \,,$$

where $\mathbf{1} \in \mathbb{R}^n$ is the all-ones vector.

#### B.1.2  Analysis of Lipschitz Continuity

For two matrices $C, C' \in \mathbb{R}^{n \times n}$, we have

$$F(C) - F(C') = \frac{1}{n}(C - C') \mathbf{1} \,.$$

Taking the 2-norm, we obtain

$$\|F(C) - F(C')\|_2 = \frac{1}{n} \|(C - C') \mathbf{1}\|_2 \,.$$

Using the inequality (the property of induced norms)

$$\|(C - C') \mathbf{1}\|_2 \le \|C - C'\|_2 \|\mathbf{1}\|_2 \,,$$

and noting that $\|\mathbf{1}\|_2 = \sqrt{n}$, we conclude:

$$\|F(C) - F(C')\|_2 \le \frac{1}{n} \|C - C'\|_2 \sqrt{n} = \frac{1}{\sqrt{n}} \|C - C'\|_2 \,.$$

Thus, $F$ is Lipschitz continuous with Lipschitz constant $L = \frac{1}{\sqrt{n}}$.

### B.1.3   Implication for Consistency

The Lipschitz continuity of $F$ implies that small perturbations in $C$ due to noise in the logit similarities result in only small changes in the extracted degree features:

$$\|F(C) - F(C')\|_2 \leq \frac{1}{\sqrt{n}}\|C - C'\|_2 .$$

Therefore, as the noise level decreases or as the number of nodes $n$ increases, thereby refining the approximation, the extracted features $F(M)$ converge to the true underlying structural descriptors of the graph.

### B.2   Part 2: Robustness of Sorting

After having proved that the row mean operator is Lipschitz continuous, we need to show that the sorting operator is stable under small perturbations, which means that if two feature vectors are close, then their sorted orders are also close.

This has been studied in the context of differentiable sorting or soft sort operators [55, 56]. The core idea behind these studies is that, assuming the extracted feature vector has distinct entries, a small perturbation that is less than half the minimum gap between any two sorted entries will leave the ordering unchanged. This means that the sorting operator is locally stable—if the input features converge, then the sorted order converges as well. As proved in Proposition 2 in [56], "*the sorting is differentiable a.e and not only converge to their hard counterparts, but also satisfy some of their properties for all $\epsilon$*". Hence, as the estimated features converge, then the sorted order converges as well.

### B.3   Part 3: Error Bound for the Approximation

Two primary sources contribute to the overall error in the approximation:

1. **Approximation Error.** The error incurred by approximating $|C_{ij} - D_{kl}|^2$ by
   $$\phi(i,k)\,\phi(j,l) = \big|f_C(i) - f_D(k)\big|\,\big|f_C(j) - f_D(l)\big|.$$
   For all indices $(i, j, k, l)$, we have proved that
   $$\epsilon_{\text{approx}} = \Big|\,|C_{ij} - D_{kl}|^2 - \phi(i,k)\phi(j,l)\Big|\gamma_{ik}\,\gamma_{jl} \;\leq\; \frac{n-1}{n^2} + \frac{m-1}{m^2}$$

2. **Sorting Stability Error.** Denote by $\epsilon_{\text{sort}}$ the error incurred by the potential change in the sorted order when the features are perturbed. Under the assumption that the true features have a positive minimal gap, the sorting operator is stable, or, in practice, one can use a soft sort with known Lipschitz properties. Hence, only small errors are induced in the sorting operator, which is bounded by small $\epsilon_{\text{sort}}$ (proved in Proposition 2 by Blondel [56]).

In summary, the error for the approximation of Gromov-Wasserstein distance results from

$$\epsilon = \epsilon_{\text{sort}} + \epsilon_{\text{approx}}$$

## C   Appendix - Graph Regularisation Bounds

We give proofs for the two Lipschitz lemmas used in Sec. C. Throughout we assume that the teacher logits are uniformly bounded $\|T\|_\infty \leq R$ , and, without loss of generality, so are the student logits $\|S\|_\infty \leq R$.

### C.1   Uniform stability

Following Bousquet and Elisseeff [57], an algorithm $\mathcal{A}$ is $\gamma$-*uniformly stable* if, for any training set $S = \{z_1, \ldots, z_n\}$, removing one example $z_i$ yields

$$\sup_z \big|\ell\big(f_{\mathcal{A}}(S), z\big) - \ell\big(f_{\mathcal{A}}(S^{\backslash i}), z\big)\big| \;\leq\; \gamma. \tag{12}$$

Uniform stability directly bounds the expected generalisation gap by $\gamma$. Bousquet and Elisseeff have also proved properties about $\gamma$:

1. If an algorithm has $\gamma$-uniform stability, then its generalization error can be controlled in $O(\gamma)$ order of magnitude.

2. $\gamma$ is approximately of the order of $L/n$, where $L$ is a Lipschitz constant of the loss function and $n$ is the number of samples.

## C.2 Lipschitz constant of GW loss

**Lemma 1** (GW Lipschitz constant). *Let $\mathcal{L}_{\mathrm{GW}}(T, S) = \lambda \, \mathrm{GW}^2(T, S)$. If $\|S\|_2 \leq R$, then*

$$\big\| \nabla_S \mathcal{L}_{\mathrm{GW}} \big\|_2 \leq \frac{64 \lambda R^3}{D} = L_{\mathrm{GW}}.$$

*where $D$ is the vocabulary size.*

*Proof.* Recall the squared Gromov–Wasserstein loss

$$\mathcal{L}_{\mathrm{GW}}(T, S) = \lambda \min_{P \in \Pi(\mu, \mu)} \sum_{i,j,k,l} \big| C_T(i,j) - C_S(k,l) \big|^2 P_{ik} P_{jl}, \qquad C_S(k,l) := (S_k - S_l)^2.$$

Let $P^\star$ denote an optimal coupling for the inner minimisation. Define the distortion tensor $D_{T,S}(k,l) = C_T(k,l) - C_S(k,l)$. For coordinate $p \in \{1, \ldots, D\}$

$$\frac{\partial \mathcal{L}_{\mathrm{GW}}}{\partial S_p} = -2\lambda \sum_{k,l} D_{T,S}(k,l) \frac{\partial C_S(k,l)}{\partial S_p} \Big[ \sum_{i,j} P_{ik}^\star P_{jl}^\star \Big].$$

Because $|S_k|, |T_k| \leq R$,

$$|D_{T,S}(k,l)| = \big| (T_k - T_l)^2 - (S_k - S_l)^2 \big| \leq 4R^2. \tag{A.1}$$

Then we consider the derivative of $C_S(k,l) = \big( S_k - S_l \big)^2$, $\qquad k, l \in \{1, \ldots, D\}$:

Taking the partial derivative with respect to the *single* logit $S_p$ gives three disjoint cases:

1. $p = k$: $\partial C_S(k,l)/\partial S_p = \partial_{S_k} (S_k - S_l)^2 = 2(S_k - S_l)$;

2. $p = l$: $\partial C_S(k,l)/\partial S_p = \partial_{S_l} (S_k - S_l)^2 = -2(S_k - S_l)$;

3. $p \neq k, l$: $C_S(k,l)$ does not depend on $S_p$, hence the derivative is 0.

All three cases are compactly encoded with Kronecker deltas:

$$\frac{\partial C_S(k,l)}{\partial S_p} = 2(S_k - S_l)\big(\delta_{kp} - \delta_{lp}\big). \tag{13}$$

where $\delta_{kp}$ and $\delta_{lp}$ denote the *Kronecker delta*:

$$\delta_{kp} = \begin{cases} 1, & k = p, \\ 0, & k \neq p, \end{cases} \qquad \delta_{lp} = \begin{cases} 1, & l = p, \\ 0, & l \neq p. \end{cases}$$

Assuming every logit is clipped within $[-R, R]$, i.e. $|S_i| \leq R$ for all $i$, we have

$$|S_k - S_l| \leq |S_k| + |S_l| \leq 2R.$$

Because $\big| \delta_{kp} - \delta_{lp} \big| \leq 1$, the absolute value of (13) is bounded by

$$\Big| \frac{\partial C_S(k,l)}{\partial S_p} \Big| \leq 2(2R) = 4R. \tag{A.2}$$

For every $(k,l)$,

$$\sum_{i,j} P_{ik}^\star P_{jl}^\star = \Big(\sum_i P_{ik}^\star\Big)\Big(\sum_j P_{jl}^\star\Big) = \mu(k)\,\mu(l) = 1/D^2. \tag{A.3}$$

Using (A.1)–(A.3) and that at most one of $\delta_{kp}, \delta_{lp}$ is non–zero,

$$\Big| \frac{\partial \mathcal{L}_{\mathrm{GW}}}{\partial S_p} \Big| \leq 2\lambda(4R^2)(4R)\sum_{k,l} \frac{1}{D^2}\big(\delta_{kp} + \delta_{lp}\big) = \frac{64 \lambda R^3}{D}.$$

Every coordinate is bounded by $\frac{64 \lambda R^3}{D}$, proving Lemma 1. $\qquad\square$

## C.3 Comparison with KL distillation (MiniED)

**Lemma 2** (KL loss Lipschitz constant). *Let $\mathcal{L}_{\mathrm{KL}}(T, S) = \lambda \, \mathrm{KL}\big(p_T \parallel p_S\big)$, $p_T = \sigma(T)$, $p_S = \sigma(S)$ with logits clipped by $\|S\|_\infty \leq R$, where the mapping $\sigma$ is the softmax function sending a logit vector $z$ to a probability vector on the $(D{-}1)$-simplex $\Delta^{D-1}$. Then*

$$\big\|\nabla_S \mathcal{L}_{\mathrm{KL}}\big\|_2 \;\leq\; \lambda \, e^R D \;=\; L_{\mathrm{KL}}. \tag{14}$$

*Proof.* By definition $\mathrm{KL}(p_T \| p_S) = \sum_i p_T(i) \big[\log p_T(i) - \log p_S(i)\big]$. Taking the derivative w.r.t. $p_S(j)$ yields

$$\frac{\partial \mathrm{KL}}{\partial p_S(j)} = -\frac{p_T(j)}{p_S(j)} \quad \Longrightarrow \quad \big|\tfrac{\partial \mathrm{KL}}{\partial p_S(j)}\big| \;\leq\; \frac{1}{\min_i p_S(i)}.$$

Because each logit is in $[-R, R]$,

$$p_S(i) = \frac{e^{S_i}}{\sum_k e^{S_k}} \;\geq\; \frac{e^{-R}}{D e^R} \;\geq\; \frac{e^{-2R}}{D}.$$

Hence $|\partial \mathrm{KL}/\partial p_S(j)| \leq e^{2R} D$.

The soft-max Jacobian $J$ satisfies $\|J\|_{2\to2} \leq \frac{1}{2}$.

$$\big\|\nabla_S \mathrm{KL}\big\|_2 \;=\; \big\|J^\top \nabla_{p_S} \mathrm{KL}\big\|_2 \;\leq\; \frac{1}{2} e^{2R} D \;\leq\; e^R D,$$

where the last inequality uses $R \geq 0$ (so $e^{2R}/2 \leq e^R$ for $R \gtrsim 1$; if $R$ is small this only tightens the bound). Multiplying by $\lambda$ proves (14). $\qquad\square$

**Comparison.** Putting the lemmas together, Lemma 2 alongside Lemma 1, we observe the *inverse* scaling of $L_{\mathrm{GW}}$ with dimension $D$:

$$L_{\mathrm{GW}} \;=\; O\!\Big(\frac{R^3}{D}\Big) \;<\; O\big(e^R D\big) = L_{\mathrm{KL}}$$

## C.4 Comparison with Wasserstein distillation (ULD)

**Lemma 3** (Lipschitz constant of the 1-Wasserstein loss). *Let $p_T = \sigma(T)$ and $p_S = \sigma(S)$ be the soft-max probabilities of teacher and student, and define*

$$\mathcal{L}_W \;=\; \lambda \, W_1\big(p_T, p_S\big),$$

*where the ground metric satisfies $d(i, j) \in [0, \Delta]$. Then*

$$\big\|\nabla_S \mathcal{L}_W\big\|_2 \;\leq\; \frac{\lambda \, \Delta \sqrt{D}}{2} \;=\; L_W.$$

*Proof.* Using Kantorovich–Rubinstein dual:

$$W_1(p_T, p_S) \;=\; \sup_{\|\varphi\|_{\mathrm{Lip}\leq 1}} \sum_i \varphi_i \big[p_T(i) - p_S(i)\big], \qquad \|\varphi\|_\infty \leq \Delta.$$

For the optimal potential $\varphi^\star$, $\nabla_{p_S} W_1 = -\varphi^\star$, hence

$$\big\|\nabla_{p_S} W_1\big\|_2 \;\leq\; \sqrt{D} \, \|\varphi^\star\|_\infty \;\leq\; \Delta\sqrt{D}.$$

The Jacobian $J_{ij} = \partial p_i / \partial S_j = p_i(\delta_{ij} - p_j)$ satisfies $\|J\|_{2\to2} \leq \frac{1}{2}$ because each row has $\ell_2$-norm $\leq \frac{1}{2}$.

$$\big\|\nabla_S W_1\big\|_2 \;=\; \|J^\top \nabla_{p_S} W_1\|_2 \;\leq\; \|J\|_{2\to2} \|\nabla_{p_S} W_1\|_2 \;\leq\; \frac{1}{2} \Delta\sqrt{D}.$$

Then multiply by the scale $\lambda$.

$$\big\|\nabla_S \mathcal{L}_W\big\|_2 \;=\; \lambda \big\|\nabla_S W_1\big\|_2 \;\leq\; \frac{\lambda \, \Delta \sqrt{D}}{2}.$$

$$\square$$

## C.5 Comparison

Combining Lemma 1, Lemma 2, and Lemma 3, we obtain the following relationship between the Lipschitz constants:

$$\frac{64\lambda R^3}{D} \quad < \quad \frac{\lambda\Delta\sqrt{D}}{2} \quad < \quad \lambda e^R D,$$

for sufficiently large vocabulary size $D$ and moderate logit range $R$.

This comparison shows that replacing the KL loss with the Gromov-Wasserstein loss results in the smallest Lipschitz constant, which implies the most stable learning process. As the Lipschitz constant is related to the uniform stability bound $\gamma \propto L/n$, a smaller Lipschitz constant translates directly to a tighter generalization gap, improving the model's ability to generalize on unseen data. Thus we can obtain the following proposition:

**Proposition 2** (Lipschitz constants comparison) We have the following relationship between the Lipschitz constants ($L$) for the Gromov-Wasserstein, Wasserstein, and KL losses:

$$L_{\text{GW}} = O\left(\frac{R^3}{D}\right) \quad < \quad L_W = O(\sqrt{D}) \quad < \quad L_{\text{KL}} = O(e^R D),$$

for sufficiently large vocabulary size $D$ and logit range $R$, which means the GW-guided training have tighter generalization bounds.

Thus, replacing GW results in the smallest Lipschitz constant, leading to better stability and tighter generalization bounds.

## C.6 Bias–variance decomposition

Following [58],

$$\mathcal{E}_{\text{gen}} \leq \underbrace{\text{Bias}(T, S)}_{\text{teacher-guided}} + \underbrace{\text{Var}}_{\leq L/n}.$$

Teacher logits fix the bias term; graph regularisation lowers the Lipschitz constant ($L_{\text{GW}} < L_{\text{KL}}$) and thus the variance term, yielding a tighter generalisation bound, especially in low-data or cross-domain settings.

# D  Appendix - Extended Experimental Setup

## D.1  Fusion Datasets details.

We construct a multi-task dataset comprising 130k examples across general reasoning, mathematics, and coding. The general data are sourced from Infinity-Instruct [36].

**Mathematics.**    The mathematical dataset we use comprises 39k pairs of math questions and corresponding answers. The questions are sourced from the $NuminaMath\_1.5$[4] dataset, while the answers are distilled from the DeepSeek-R1-671B model by the AM team[5]. $NuminaMath\_1.5$ represents the second iteration of the widely acclaimed NuminaMath[37] dataset. It offers high-quality data for competition-level mathematical problems across various domains, including Algebra, Geometry, Combinatorics, Calculus, Inequalities, Logic and Puzzles, and Number Theory.

**Coding.**    In the code generation domain, we utilized the KodCode-V1-SFT-R1 dataset [38], which contains approximately 268k samples. Each sample was fed into our pivot model, which was prompted to generate 5 random responses per input. The generated outputs were then passed through a sandbox-based evaluation system. For each sample, if at least one of the five responses failed, the sample was flagged for further consideration. From these flagged samples, we filtered and selected 39k high-quality examples for further distillation.

---

[4]https://huggingface.co/datasets/AI-MO/NuminaMath-1.5
[5]https://huggingface.co/datasets/a-m-team/AM-DeepSeek-R1-Distilled-1.4M

**Training Details.** We train with C-AdamW and cosine decay for 5 epochs, using early stopping on the 4th. Fusion is performed with offline-extracted hidden states (before the `lm_head`) to reduce GPU cost, requiring 1.5TB of storage. Fusion uses a batch size of 16 on 8×80GB NVIDIA H800 GPUs for 20 hours. We adopt early stopping at epoch 4 (of 5), learning rate $1 \times 10^{-6}$, and default ULD weight $\lambda = 0.5$, GLD weight $\lambda = 0.001$.

**Offline Teacher Loading.** For InfiGFusion, we extract hidden states (before the `lm_head`) from source models and store them ( 1.5TB per run). Only final layers are retained online, reducing memory and speeding up training.

# E Appendix - Evaluation Prompt

To enhance the robustness of answer extraction under the regex-based evaluation framework of OpenCompass [59] and EvalPlus [60], we systematically refine the prompts used in several benchmark datasets. These tailored prompt formats are designed to facilitate precise output matching, mitigating ambiguities that often arise from model generations. The revised prompt templates corresponding to each dataset are presented in Table 7, which details how task instructions and answer formats are standardized to align with OpenCompass's automatic evaluation pipeline.

For datasets such as TheoremQA and HumanEval, we retain the original prompt configurations, adhering to their respective community-adopted evaluation protocols. This ensures consistency with prior works and preserves the validity of established benchmarks. For MBPP, we utilize EvalPlus [60] for more rigorous assessment of LLM-generated code, providing enhanced reliability in functional correctness evaluation.

Table 7: Prompt format used for different datasets.

| Dataset | Prompt Format |
|---|---|
| IFEval | `{prompt}\nPlease directly give the correct answer:` |
| ARC-C | `Question: {question}\nA . {textA}\nB . {textB}\nC . {textC}\nD . {textD}\nDirectly give me the correct answer option, and then explain:` |
| Hellaswag | `{ctx}\nQuestion : Which ending makes the most sense?\nDirectly give me the correct choice, you can further explain it or not.\nA . {A}\nB . {B}\nC . {C}\nD . {D}\nYou may choose from 'A', 'B', 'C', 'D'.\nAnswer :` |
| BBH | `Follow the given examples and answer the question.\n {_hint}\nQ : {{input}}\nA : Let's think step by step.` |
| DROP | `You will be asked to read a passage and answer a question. Some examples of passages and Q\&A are provided below.\n {drop_examples}\n \n ## Your Task\n --\n {prompt}\nThink step by step, then write a line of the form "Answer: \$ANSWER" at the end of your response.` |
| MMLU | `{_hint}\nQuestion : {{input}}\nA . {{A}}\nB . {{B}}\nC . {{C}}\nD . {{D}}\n \nFor simple problems:\nDirectly provide the answer with minimal explanation.\n \nFor complex problems:\nUse this step-by-step format:\n #### Step 1: [Concise description]\n [Brief explanation]\n #### Step 2: [Concise description]\n [Brief explanation]\n \nRegardless of the approach, always conclude with:\nThe answer is [the\_answer\_letter].\nwhere the [the\_answer\_letter] is one of A, B, C or D.\n \nLet 's think step by step.` |
| GSM8K | `{question}\nPlease reason step by step, and put your final answer within \\boxed\{\}.` |
| MATH | `{problem}\nPlease reason step by step, and put your final answer within \\boxed\{\}.` |

Table 8: Performance on BBH reasoning tasks. Tasks marked in **bold** involve complex relational reasoning, multi-hop inference, or semantic alignment, where InfiGFusion shows notable improvements over token-level fusion baselines.

| BBH Dataset | Qwen2.5 | Mistral-24B | Phi4-14B | DeepSeek | Phi4-14B (SFT) | InfiGFusion |
|---|---|---|---|---|---|---|
| bbh-temporal_sequences | 98.8 | 98.8 | 99.6 | 66.8 | 98.8 | 100 |
| bbh-disambiguation_qa | 78.8 | 75.6 | 82.4 | 74.8 | 79.6 | 76 |
| bbh-date_understanding | 90.4 | 86.8 | 94.8 | 50.4 | 85.2 | 82 |
| bbh-tracking_shuffled_objects_three_objects | 86.8 | 99.2 | 99.6 | 75.6 | 97.2 | 99.6 |
| bbh-penguins_in_a_table | 90.41 | 97.95 | 98.63 | 69.86 | 93.84 | 97.26 |
| bbh-geometric_shapes | 58.8 | 49.2 | 56.4 | 71.6 | 56.4 | 62.4 |
| bbh-snarks | 84.27 | 85.96 | 61.8 | 47.19 | 70.22 | 79.21 |
| bbh-ruin_names | 82.8 | 76.8 | 88.8 | 39.6 | 88 | 88.4 |
| bbh-tracking_shuffled_objects_seven_objects | 80.4 | 96.4 | 94.4 | 80.8 | 90 | 96.8 |
| bbh-tracking_shuffled_objects_five_objects | 79.2 | 99.2 | 96.8 | 82.8 | 95.6 | 96.8 |
| bbh-logical_deduction_three_objects | 97.6 | 98.8 | 98.4 | 83.2 | 96.4 | 97.6 |
| bbh-hyperbaton | 97.2 | 98.8 | 49.2 | 52 | 53.2 | 99.6 |
| bbh-logical_deduction_five_objects | 80.4 | 82 | 85.6 | 86 | 92.4 | 94 |
| bbh-logical_deduction_seven_objects | 68.8 | 62.8 | 88.4 | 83.2 | 88.8 | 89.2 |
| bbh-movie_recommendation | 72.8 | 78 | 58.8 | 60.8 | 62.8 | 43.6 |
| bbh-salient_translation_error_detection | 70.4 | 63.2 | 65.2 | 54.4 | 64 | 67.2 |
| bbh-reasoning_about_colored_objects | 93.6 | 90 | 96.4 | 87.6 | 96.8 | 96.4 |
| bbh-multistep_arithmetic_two | 96.4 | 93.2 | 64 | 81.6 | 62 | 99.6 |
| bbh-navigate | 97.2 | 80.8 | 96 | 51.6 | 82 | 96 |
| bbh-dyck_languages | 35.6 | 37.2 | 11.2 | 9.6 | 13.2 | 40 |
| bbh-word_sorting | 32 | 36.4 | 0.4 | 0 | 9.6 | 59.2 |
| bbh-sports_understanding | 79.2 | 91.2 | 72.4 | 2 | 89.2 | 92.4 |
| bbh-boolean_expressions | 55.6 | 87.2 | 28.8 | 8 | 66.4 | 98.4 |
| bbh-object_counting | 89.6 | 95.2 | 98.8 | 99.6 | 99.6 | 98 |
| bbh-formal_fallacies | 54.8 | 73.2 | 0.4 | 0.8 | 0.8 | 91.6 |
| bbh-causal_judgement | 43.85 | 68.98 | 32.99 | 45.35 | 35.83 | 70.05 |
| bbh-web_of_lies | 99.2 | 100 | 100 | 93.6 | 100 | 100 |

# F    Appendix - Comprehensive Reasoning Performance

## F.1    Analysis on BBH: Semantic Dependency in Complex Reasoning

We conduct a comprehensive evaluation of InfiGFusion on the BBH benchmark, which consists of diverse and challenging reasoning tasks requiring multi-step logic, relational alignment, and semantic consistency. These tasks pose significant challenges for token-level fusion methods due to their reliance on token co-activation patterns and internal reasoning structures.

*Remark.* The use of "internal reasoning" refers to the relational structure among semantic dimensions in the logit space. We align inter-logit relation graphs via Gromov–Wasserstein, providing a structural inductive bias for reasoning-heavy tasks, without explicitly modeling the reasoning steps.

**Semantic Dependency Fusion Matters.**    InfiGFusion demonstrates clear advantages on tasks where reasoning relies on inter-token dependencies rather than isolated token probabilities. For example, in **Logical Deduction (5 objects / 7 objects)**, InfiGFusion outperforms the SFT baseline by +1.6% and +0.4%, respectively, indicating better preservation of relational structures. In **Tracking Shuffled Objects** tasks (five and seven objects), InfiGFusion matches or exceeds the best-performing models, showing that graph-based alignment can maintain structural consistency in object tracking.

**Superior Multi-hop and Arithmetic Reasoning.**    Tasks such as **Multistep Arithmetic Two** (+36.8% over SFT) and **Causal Judgement** (+31.55%) highlight InfiGFusion's strength in capturing multi-hop reasoning and causal dependencies, which are notoriously difficult for token-level methods. These improvements suggest that GLD loss effectively aligns higher-order reasoning structures across source models.

**Recovering from Source Model Weaknesses.**    InfiGFusion mitigates the weaknesses of individual source models in cases where token-level fusion underperforms. For instance, in **Boolean Expressions**, InfiGFusion achieves 98.4%, significantly outperforming phi4 SFT (66.4%) and other baselines,

by aligning semantic relationships between logical operators through graph structures. Similarly, in **Dyck Languages**—a task requiring bracket matching and structural hierarchy—InfiGFusion closes the gap (+23.6% over SFT), showcasing its advantage in structure-sensitive tasks.

**Robustness in Semantic Alignment.** On tasks like **Ruin Names** and **Reasoning about Colored Objects**, InfiGFusion maintains top-tier performance, matching or slightly exceeding SFT results. This indicates that InfiGFusion not only aggregates external knowledge but also preserves the pivot model's strengths, avoiding catastrophic forgetting.

**Challenges and Observations.** While InfiGFusion excels in structure-dependent reasoning, it shows less improvement in tasks dominated by shallow lexical patterns (e.g., **Word Sorting**, **Sports Understanding**), where token-level matching suffices. This suggests that InfiGFusion's strength lies in modeling inter-token dependencies rather than surface-level token alignment.

Overall, InfiGFusion outperforms token-level fusion methods on BBH tasks requiring structural reasoning, multi-step inference, and semantic consistency. By explicitly modeling logit-space semantic dependencies, InfiGFusion enables robust and interpretable fusion of heterogeneous LLMs.

## F.2 Analysis on MMLU: Fusion of Knowledge and Reasoning Behaviors

We further evaluate InfiGFusion on the MMLU benchmark, covering 57 sub-domains across STEM, social sciences, law, and humanities. Unlike BBH, which emphasizes complex reasoning chains, MMLU focuses on factual knowledge, domain-specific reasoning, and multi-field coverage, offering a complementary perspective to assess fusion effectiveness.

**Retaining Pivot Knowledge, Enhancing Structural Alignment.** InfiGFusion demonstrates strong consistency with the pivot model's strengths, while further integrating structural reasoning from source models. On factual knowledge tasks like **High School Computer Science** (95%) and **US History** (91.67%), InfiGFusion matches or slightly improves over Phi4 SFT. This indicates that structure-aware fusion preserves core pivot capabilities without degradation.

**Advantages in Structure-sensitive Tasks.** InfiGFusion outperforms Phi4 SFT on structure-dependent categories such as **Abstract Algebra** (+2%) and **Security Studies** (+1.63%), confirming its effectiveness in aligning semantic dependencies. Similarly, in tasks like **Machine Learning** and **Computer Security**, InfiGFusion maintains competitive results, highlighting its ability to integrate relational knowledge from multiple sources.

**Balanced Performance in STEM-heavy Tasks.** While DeepSeek-R1 exhibits domain-specific advantages (e.g., **College Mathematics**: 93%), InfiGFusion remains competitive (81%), achieving a balance between generalization and specialized knowledge. On **Clinical Knowledge** and **Medical Genetics**, InfiGFusion slightly trails SFT but still improves upon the raw Phi4 model, reflecting its capacity to enhance reasoning fidelity without overfitting to narrow domains.

**Stable Gains in Social Sciences and Law.** In domains like **International Law**, **Global Facts**, and **World Religions**, InfiGFusion maintains accuracy on par with SFT, demonstrating robustness across disciplines. Notably, in **Sociology**, InfiGFusion surpasses all baselines (90.05%), suggesting effective fusion of nuanced reasoning patterns.

InfiGFusion's gains are most prominent in tasks requiring structural reasoning and multi-source knowledge integration, while its performance on pure factual recall tasks remains on par with SFT. This validates our design choice of graph-based semantic alignment for model fusion, enabling balanced and interpretable improvements.

# G    Appendix - Relative Error of GW

**Setup.** To better analyze the relative error of the proposed GW, we quantify the *scale* of the GW distance and its *relative error* on representative BBH subsets using a 14B student and $\sim$130K training examples. We compare three objectives: (i) our *sorting-based* closed-form approximation ($\mathcal{O}(n \log n)$), (ii) *Sinkhorn GW* ($\mathcal{O}(n^2 \log n)$), and (iii) *Entropic GW* ($\mathcal{O}(n^2 \log n)$). Following [61,

Table 9: Performance on MMLU sub-tasks. InfiGFusion excels in structure-sensitive reasoning while maintaining robust general knowledge performance. Bold categories indicate tasks where structural dependencies and multi-source knowledge integration are critical.

| MMLU Dataset | Qwen2.5 | Mistral-24B | Phi4-14B | DeepSeek | Phi4-14B (SFT) | InfiGFusion |
|---|---|---|---|---|---|---|
| lukaemon_mmlu_college_biology | 87.5 | 90.97 | 96.53 | 94.44 | 94.44 | 92.36 |
| lukaemon_mmlu_college_chemistry | 58 | 66 | 72 | 64 | 66 | 63 |
| lukaemon_mmlu_college_computer_science | 76 | 78 | 86 | 82 | 82 | 85 |
| lukaemon_mmlu_college_mathematics | 80 | 72 | 79 | 93 | 77 | 81 |
| lukaemon_mmlu_college_physics | 78.43 | 89.22 | 94.12 | 92.16 | 88.24 | 84.31 |
| lukaemon_mmlu_electrical_engineering | 77.93 | 78.62 | 82.07 | 80.69 | 82.07 | 82.76 |
| lukaemon_mmlu_astronomy | 87.5 | 94.08 | 90.79 | 92.76 | 89.47 | 90.79 |
| lukaemon_mmlu_anatomy | 76.3 | 79.26 | 82.96 | 74.81 | 80.74 | 78.52 |
| lukaemon_mmlu_abstract_algebra | 73 | 65 | 82 | 85 | 86 | 88 |
| lukaemon_mmlu_machine_learning | 67.86 | 70.54 | 80.36 | 77.68 | 81.25 | 78.57 |
| lukaemon_mmlu_clinical_knowledge | 84.15 | 87.17 | 84.91 | 83.77 | 89.43 | 87.55 |
| lukaemon_mmlu_global_facts | 53 | 62 | 55 | 57 | 53 | 53 |
| lukaemon_mmlu_management | 86.41 | 87.38 | 85.44 | 86.41 | 84.47 | 87.38 |
| lukaemon_mmlu_nutrition | 82.68 | 85.95 | 87.25 | 84.97 | 87.91 | 84.31 |
| lukaemon_mmlu_marketing | 90.17 | 92.31 | 92.74 | 89.32 | 92.74 | 95.3 |
| lukaemon_mmlu_professional_accounting | 75.89 | 73.4 | 86.17 | 78.72 | 84.75 | 81.91 |
| lukaemon_mmlu_high_school_geography | 87.88 | 87.88 | 90.4 | 88.89 | 91.92 | 89.9 |
| lukaemon_mmlu_international_law | 81.82 | 81.82 | 91.74 | 82.64 | 90.91 | 90.91 |
| lukaemon_mmlu_moral_scenarios | 71.96 | 68.6 | 75.75 | 73.41 | 74.75 | 73.97 |
| lukaemon_mmlu_computer_security | 81 | 85 | 85 | 82 | 83 | 87 |
| lukaemon_mmlu_high_school_microeconomics | 88.24 | 90.34 | 95.38 | 94.96 | 96.22 | 96.22 |
| lukaemon_mmlu_professional_law | 56.65 | 61.73 | 67.28 | 61.67 | 64.99 | 65.71 |
| lukaemon_mmlu_medical_genetics | 89 | 89 | 93 | 94 | 92 | 91 |
| lukaemon_mmlu_professional_psychology | 80.07 | 82.19 | 85.78 | 79.58 | 86.6 | 84.97 |
| lukaemon_mmlu_jurisprudence | 82.41 | 85.19 | 92.59 | 86.11 | 86.11 | 86.11 |
| lukaemon_mmlu_world_religions | 84.21 | 88.89 | 90.06 | 90.64 | 89.47 | 88.3 |
| lukaemon_mmlu_philosophy | 79.74 | 80.06 | 84.89 | 77.17 | 83.92 | 82.96 |
| lukaemon_mmlu_virology | 52.41 | 49.4 | 53.01 | 54.22 | 53.61 | 54.22 |
| lukaemon_mmlu_high_school_chemistry | 76.85 | 83.25 | 90.15 | 88.67 | 88.67 | 84.24 |
| lukaemon_mmlu_public_relations | 70.91 | 69.09 | 77.27 | 74.55 | 79.09 | 75.45 |
| lukaemon_mmlu_high_school_macroeconomics | 84.87 | 84.1 | 89.74 | 88.72 | 90.51 | 91.03 |
| lukaemon_mmlu_human_sexuality | 83.21 | 88.55 | 86.26 | 86.26 | 87.79 | 85.5 |
| lukaemon_mmlu_elementary_mathematics | 95.24 | 94.97 | 97.09 | 97.35 | 96.83 | 95.24 |
| lukaemon_mmlu_high_school_physics | 78.81 | 76.82 | 85.43 | 84.11 | 87.42 | 84.11 |
| lukaemon_mmlu_high_school_computer_science | 92 | 92 | 96 | 93 | 93 | 95 |
| lukaemon_mmlu_high_school_european_history | 81.82 | 80 | 84.24 | 84.85 | 83.03 | 83.64 |
| lukaemon_mmlu_business_ethics | 75 | 83 | 82 | 82 | 82 | 82 |
| lukaemon_mmlu_moral_disputes | 76.88 | 78.61 | 81.79 | 74.57 | 83.24 | 81.5 |
| lukaemon_mmlu_high_school_statistics | 77.78 | 80.56 | 88.89 | 89.81 | 83.8 | 88.43 |
| lukaemon_mmlu_miscellaneous | 91.57 | 94.25 | 93.87 | 92.46 | 93.74 | 91.7 |
| lukaemon_mmlu_formal_logic | 68.25 | 66.67 | 77.78 | 91.27 | 78.57 | 74.6 |
| lukaemon_mmlu_high_school_government_and_politics | 93.78 | 95.34 | 95.85 | 96.37 | 96.89 | 96.37 |
| lukaemon_mmlu_prehistory | 86.11 | 83.64 | 85.8 | 83.64 | 87.35 | 87.65 |
| lukaemon_mmlu_security_studies | 77.96 | 78.78 | 77.14 | 78.37 | 79.59 | 81.22 |
| lukaemon_mmlu_high_school_biology | 90 | 90.97 | 94.19 | 92.58 | 92.9 | 94.19 |
| lukaemon_mmlu_logical_fallacies | 85.28 | 84.66 | 86.5 | 85.28 | 87.73 | 87.73 |
| lukaemon_mmlu_high_school_world_history | 87.76 | 88.61 | 89.87 | 89.45 | 88.61 | 89.87 |
| lukaemon_mmlu_professional_medicine | 84.19 | 92.28 | 91.18 | 84.19 | 92.28 | 91.54 |
| lukaemon_mmlu_high_school_mathematics | 85.93 | 84.07 | 92.22 | 97.04 | 90.37 | 90 |
| lukaemon_mmlu_college_medicine | 79.77 | 83.82 | 84.97 | 88.44 | 85.55 | 85.55 |
| lukaemon_mmlu_high_school_us_history | 89.71 | 86.76 | 92.16 | 87.75 | 91.67 | 91.67 |
| lukaemon_mmlu_sociology | 86.57 | 85.57 | 89.05 | 84.58 | 88.06 | 90.05 |
| lukaemon_mmlu_econometrics | 60.53 | 64.91 | 71.93 | 65.79 | 71.93 | 67.54 |
| lukaemon_mmlu_high_school_psychology | 90.64 | 92.84 | 94.31 | 91.38 | 94.86 | 95.6 |
| lukaemon_mmlu_human_aging | 73.99 | 75.34 | 80.72 | 78.48 | 78.48 | 78.92 |
| lukaemon_mmlu_us_foreign_policy | 89 | 88 | 92 | 88 | 93 | 91 |
| lukaemon_mmlu_conceptual_physics | 88.09 | 86.81 | 89.79 | 91.06 | 90.64 | 88.09 |

| Method | Logical Deduction | Multistep Arithmetic | Causal Judgement | Dyck Languages |
|---|---|---|---|---|
| *Our Approximation (Sorting-Based, $\mathcal{O}(n\log n)$)* | | | | |
| GW Distance | 0.0356 | 0.0614 | 0.0445 | 0.0210 |
| Abs. Error Bound | 0.00015 | 0.00015 | 0.00015 | 0.00015 |
| Relative Error (%) | 0.421 | 0.244 | 0.337 | 0.714 |
| Performance (%) | 89.2 | 99.6 | 70.0 | 94.4 |
| *Sinkhorn GW ($\mathcal{O}(n^2\log n)$)* | | | | |
| GW Distance | 0.02145 | 0.0618 | 0.0449 | 0.0213 |
| Abs. Error Bound | 0.00009 | 0.00009 | 0.00009 | 0.00009 |
| Relative Error (%) | 0.419 | 0.145 | 0.200 | 0.422 |
| Performance (%) | 89.9 | 99.2 | 70.4 | 94.1 |
| *Entropic GW ($\mathcal{O}(n^2\log n)$)* | | | | |
| GW Distance | 0.0314 | 0.0617 | 0.0448 | 0.0212 |
| Abs. Error Bound | 0.00012 | 0.00012 | 0.00012 | 0.00012 |
| Relative Error (%) | 0.382 | 0.194 | 0.267 | 0.566 |
| Performance (%) | 89.7 | 99.6 | 70.7 | 93.9 |

Table 10: Scale of GW distance and relative error on BBH subsets (14B model, $\sim$130K data).

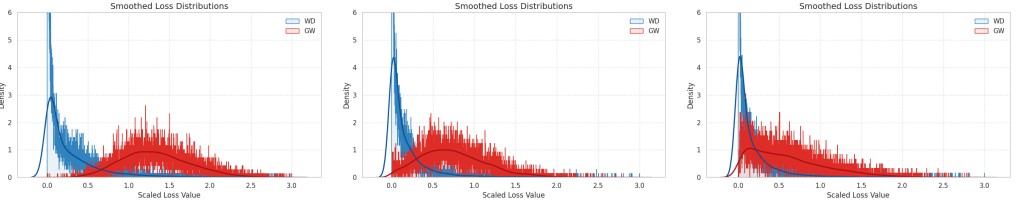

Figure 5: Comparison of WD and GW distributions during fusion. Left: before distillation; Middle: after token-level WD optimization. WD reduces significantly, but GWD remains largely unchanged, indicating semantic dependency misalignment; Right: after GLD optimization.

62], we report a certified *absolute error bound* and compute *relative error* as $\mathrm{RE} = \frac{\text{Abs. Error Bound}}{\text{GW Distance}} \times 100\%$. We also list the end-task performance (%) obtained when optimizing each objective.

**Findings.** (1) *Distance scale and RE.* GW distances on BBH fall between 0.02 and 0.06. With the certified absolute bounds, relative errors are consistently below $1\%$ (Table 10), supporting the suitability of using the approximation as a training objective. (2) *Approximation quality vs. cost.* Sinkhorn/Entropic GW yield slightly tighter RE than the sorting-based objective, but with $\mathcal{O}(n^2\log n)$ cost. Our sorting-based objective offers near-parity performance while remaining $\mathcal{O}(n\log n)$ and thus *scalable*. (3) *Empirical consistency.* The sorting-based absolute error aligns with the expected $\mathcal{O}(1/n)$ rate at our operating regimes (e.g., $n \approx 150\text{K}$, $m \approx 100\text{K}$), while competitive end-task performance validates training usefulness.

**On larger models and time gaps.** The section 4.7, Table 10, and expanded distillation experiments with Table 12 give further discussion. While on smaller students the wall-clock gap between Sinkhorn and our sorting-based objective can appear modest, the *asymptotic* gap widens with scale. Concretely, the effective problem size $n$ grows with sequence length, batch size, and active logit support (e.g., top-$k$ neighbors), making Sinkhorn/Entropic $\mathcal{O}(n^2\log n)$ costs rise superlinearly, whereas our objective scales as $\mathcal{O}(n\log n)$. Empirically, on larger models with more dimensionality, we observe proportionally larger savings in GW-side time while maintaining comparable downstream performance.

At the scales of interest for LLM fusion, GW distances are small but stable; certified relative errors under our approximation remain $< 1\%$, and the $\mathcal{O}(n\log n)$ objective delivers the right trade-off between fidelity and throughput, with advantages that grow with model/sequence scale.

| Condition-dependent Logic (Object Tracking) | |
|---|---|
| **Task Type: Sequential state updates under conditions** | |
| **InfiGFusion Behavior** | **Baseline Behavior** |

"prompt":  "Follow the given examples and answer the question.\nAnswer extremely simple questions about the colors of objects on a surface.\n\nQ: On the nightstand, there is a red pencil, a purple mug, a burgundy keychain, a fuchsia teddy bear, a black plate, and a blue stress ball.  ......  \n\nQ: On the floor, you see a green bracelet, a purple cat toy, a brown pair of sunglasses, a black fidget spinner, a red dog leash, and an orange pen.  How many objects are neither black nor blue?\nOptions:\n(A) zero\n(B) one\n(C) two\n(D) three\n(E) four\n(F) five\n(G) six\nA: Let's think step by step.\n"

| "origin_prediction":  "According to this question, the objects on the floor are as follows:  a green bracelet, a purple cat toy, a brown pair of sunglasses, a black fidget spinner, a red dog leash, and an orange pen.\n\nWe need to count how many objects are neither black nor blue.\n\n1.  Green bracelet:  Not black or blue → count\n2.  Purple cat toy:  Not black or blue → count\n3.  Brown pair of sunglasses:  Not black or blue → count\n4.  Black fidget spinner:  Black → do not count\n5.  Red dog leash:  Not black or blue → count\n6.  Orange pen:  Not black or blue → count\n\nSo, the objects that are neither black nor blue are:  green bracelet, purple cat toy, brown pair of sunglasses, red dog leash, and orange pen.  That makes 5 objects.\n\nThe answer is (F). ", "correct":  true | "origin_prediction":  "Let's analyze the question step by step:\n\nThe objects on the floor are:\n1.  A green bracelet\n2.  A purple cat toy\n3.  A brown pair of sunglasses\n4.  A black fidget spinner\n5.  A red dog leash\n6.  An orange pen\n\nWe need to determine how many objects are neither black nor blue.\n\n- The green bracelet is neither black nor blue.\n- The purple cat toy is neither black nor blue.\n- The brown pair of sunglasses is neither black nor blue.\n- The black fidget spinner is black, so it is excluded.\n- The red dog leash is neither black nor blue.\n- The orange pen is neither black nor blue.\n\nCounting the objects that are neither black nor blue, we have:\n1.  Green bracelet\n2.  Purple cat toy\n3.  Brown pair of sunglasses\n4.  Red dog leash\n5.  Orange pen\n\nThere are 5 objects that are neither black nor blue.\n\nTherefore, the answer is (F) five.", "correct":  true |

Table 11: Case study on condition-dependent object tracking logic.

## H  Appendix - Distribution Analysis of WD and GW During Fusion

To better understand how InfiGFusion improves semantic alignment, we visualize the evolution of Wasserstein Distance (WD) and Gromov-Wasserstein Distance (GWD) distributions across three key stages of fusion (Fig. 5).

**Experimental Setup.** We compare: (1) The initial distribution before distillation (Left), (2) After token-level WD optimization using ULD loss (Middle), (3) After full GLD+ULD optimization (Right). Each histogram reflects pairwise distances between pivot and source logits across sampled instances.

**Why This Matters:** WD measures the marginal alignment of token logits, while GWD captures relational (dependency-level) differences between token dimensions. Existing fusion methods (e.g., ULD) optimize WD but ignore semantic dependencies, making GWD a critical metric for evaluating structural alignment.

Before distillation (Left): Both WD and GWD exhibit large values, indicating poor alignment at both token-level and structure-level. After ULD optimization (Middle): WD reduces significantly, showing that token-wise distributions are aligned.  However, GWD remains largely unchanged, revealing a persistent mismatch in semantic dependency patterns.  This validates our hypothesis: token-level objectives fail to capture cross-token interactions essential for reasoning consistency. After GLD+ULD optimization (Right): Both WD and GWD distributions shift leftward. Notably,

GWD exhibits a clear reduction, demonstrating that GLD successfully aligns relational structures in logit space, complementing token-level alignment.

# I    Appendix - Case study

We provide additional results compared between InfiGFusion and Phi-4 SFT in Table.11.

# J    Appendix - Additional Discussion

Our approach focuses on *relational consistency*, aligning the semantic structure of logits across sources, without an explicit mechanism for resolving *factual conflicts* between models. In practice, when source models encode contradictory knowledge, fusion can produce outputs that are structurally coherent yet factually incorrect. While the framework naturally accommodates extensions such as confidence estimation and external verifiers, our preliminary trials with entropy-based source selection [13] and Bayesian conditional alignment [63] yielded only marginal gains at our scales and benchmarks.

We view conflict-aware *source re-weighting and selection* as complementary to GLD rather than a substitute. Promising directions include per-claim uncertainty calibration, verifier-in-the-loop pipelines (e.g., retrieval or factuality checks), and adaptive top-$k$ pruning conditioned on detected conflicts. A principled, high-yield conflict-resolution module is thus an important avenue for future work, whereas this paper concentrates on semantic structure alignment and a stronger GLD objective.

# K    Appendix - Additional Distillation Experiments

To verify the generality of our Graph-on-Logits Distillation (GLD) loss beyond model fusion, we conduct additional experiments in the traditional knowledge distillation setting, where a compact student model learns from a single larger teacher model. These experiments assess whether GLD can also improve response quality under extreme model compression (from 7B to sub-1B scale).

**Distillation Datasets:** The **distillation** performance of GLD is evaluated via instruction-following accuracy averaged over five random seeds. For distillation, we follow prior work and use Dolly-15k [64] for training. We follow [65] that evaluate on four benchmarks: SelfInst [66], VicunaEval [67], Super Natural Instructions (S-NI) [68], and the Dolly [64].

**Models:** For distillation, we distill from LLaMA3-8B, Mistral-7B, and Qwen2.5-7B into student models, including GPT2-120M [69], OPT-350M [70], and Bloomz-560M [71].

**Training settings:** Distillation uses LoRA finetuning and temperature $\tau = 1.5$. All models use a learning rate of $1 \times 10^{-6}$ and $\lambda = 0.5$ unless otherwise stated. For all experiments, evaluation metrics are averaged across 5 random seeds.

Table 12 presents a comprehensive comparison of GLD against prior distillation baselines across a wide range of teacher–student pairs. Across all 27 configurations (3 teachers × 3 students × 3 methods), GLD achieves the highest average accuracy in nearly all cases, consistently outperforming both MinED and ULD.

Notably, even when distilling into extremely lightweight students such as GPT2-120M, GLD shows robust generalization on challenging reasoning datasets like SelfInst and VicunaEval. For instance, when distilled from Mistral to GPT2, GLD yields an average gain of +1.39 over ULD and +1.39 over MinED, despite requiring fewer GPU hours than ULD. Similarly, from LLaMA3 to OPT-350M, GLD improves +0.53 over the best baseline with competitive efficiency.

This performance shows that GLD's structure-aware objective can preserves inter-token relationships critical for multi-step and relational inference, particularly impactful for small-capacity models where token-level alignment alone often fails. Compared to ULD, which aligns logits dimension-wise using approximated Wasserstein distance, GLD additionally distills cross-dimensional co-activations via logit graphs, serving as a richer supervisory signal. Its effectiveness across diverse teachers (Qwen, LLaMA3, Mistral) and architectures (GPT-style, OPT, Bloomz) highlights GLD's compatibility with heterogeneous generation styles and token spaces.

Table 12: Performance of different distillation methods across teacher-student pairs. Rouge-L is averaged over 5 different seeds on 4 instruction-following benchmarks. Bold indicates the best performance. **Improv.** denotes the relative performance gain of GLD over the other baseline.

| Teacher | Model | Methods | Times/h | Dolly | SelfInst | VicunaEval | S-NI | Average | Improv. |
|---|---|---|---|---|---|---|---|---|---|
| Teacher | Qwen 2.5-7B | | | 28.71±0.31 | 26.01±0.43 | 21.32±0.48 | 42.56±0.32 | 29.65±0.39 | |
| | LLama3 | SFT | | 32.05±0.38 | 25.04±0.71 | 20.68±0.47 | 40.73±0.29 | 29.63±0.46 | |
| | Mistral | | | 31.46±0.34 | 25.15±0.73 | 20.57±0.29 | 37.56±0.40 | 28.69±0.44 | |
| Qwen 2.5 | GPT2-120M | MinED | 6.5 | 20.55±0.15 | 10.88±0.36 | 14.90±0.63 | 22.12±0.36 | 17.11±0.38 | 3.48% |
| | | ULD | 2.5 | 20.88±0.56 | 11.02±0.33 | 15.45±0.36 | 21.24±0.37 | 17.15±0.41 | 3.27% |
| | | GLD | 3.8 | 21.38±0.38 | **11.34±0.42** | **15.73±0.15** | **22.38±0.30** | **17.58±0.31** | - |
| | OPT-350m | MinED | 5.0 | 22.62±0.34 | 12.37±0.45 | 15.52±0.30 | 21.99±0.12 | 18.13±0.30 | 0.73% |
| | | ULD | 4.5 | 22.36±0.29 | 11.38±0.50 | 14.61±0.36 | 21.80±0.59 | 17.54±0.44 | 4.11% |
| | | GLD | 5.0 | 22.80±0.48 | **12.61±0.54** | **15.77±0.67** | 21.85±0.33 | **18.26±0.51** | - |
| | Bloomz-560m | MinED | 4.2 | **22.86±0.56** | 11.78±0.34 | **14.82±0.39** | 22.83±0.30 | 18.07±0.40 | 1.37% |
| | | ULD | 5.2 | 22.51±0.32 | 12.48±0.44 | 14.08±0.45 | 22.82±0.21 | 17.97±0.36 | 1.93% |
| | | GLD | 4.9 | 22.83±0.16 | **12.52±0.14** | 14.74±0.31 | 23.19±0.13 | **18.32±0.19** | - |
| LLama3 | GPT2-120M | MinED | 2.9 | 19.93±0.36 | 10.15±0.24 | 14.38±0.43 | 19.78±0.23 | 16.06±0.32 | 3.44% |
| | | ULD | 2.5 | 19.66±0.49 | 9.84±0.31 | 14.59±0.50 | 19.77±0.14 | 15.97±0.36 | 4.06% |
| | | GLD | 2.8 | **20.97±0.28** | **10.81±0.55** | 14.97±0.37 | **20.09±0.24** | **16.69±0.36** | - |
| | OPT-350m | MinED | 5.3 | 21.25±0.27 | 10.06±0.38 | **15.28±0.41** | 19.04±0.32 | 16.41±0.35 | 2.18% |
| | | ULD | 4.4 | 21.08±0.56 | 9.85±0.42 | 14.89±0.50 | 19.13±0.58 | 16.24±0.52 | 3.25% |
| | | GLD | 4.7 | **22.01±0.31** | **10.97±0.33** | 14.87±0.46 | 19.21±0.25 | **16.77±0.34** | - |
| | Bloomz-560m | MinED | 4.7 | 20.16±0.21 | 10.81±0.16 | 13.36±0.44 | 20.62±0.17 | 16.24±0.25 | 2.97% |
| | | ULD | 5.3 | 20.69±0.49 | 10.50±0.64 | 12.97±0.61 | 20.71±0.18 | 16.22±0.48 | 3.10% |
| | | GLD | 5.5 | **21.07±0.47** | 10.82±0.17 | 13.41±0.29 | 21.58±0.24 | **16.72±0.29** | - |
| Mistral | GPT2-120M | MinED | 2.9 | 20.47±0.62 | 10.52±0.79 | 15.36±0.27 | 20.59±0.54 | 16.74±0.56 | 8.34% |
| | | ULD | 2.6 | 20.00±0.52 | 10.53±0.81 | 15.29±0.20 | 20.57±0.16 | 16.60±0.42 | 9.23% |
| | | GLD | 2.6 | **21.90±0.31** | **11.89±0.41** | 15.99±0.31 | **22.74±0.27** | **18.13±0.33** | - |
| | OPT-350m | MinED | 3.1 | 22.67±0.29 | 11.96±0.74 | 15.85±0.65 | 22.55±0.20 | 18.26±0.47 | 3.63% |
| | | ULD | 2.3 | 22.45±0.38 | 11.19±0.88 | 15.41±0.60 | 23.17±0.32 | 18.06±0.55 | 4.79% |
| | | GLD | 2.8 | **23.72±0.36** | **12.73±0.30** | 15.14±0.53 | **24.11±0.28** | **18.92±0.36** | - |
| | Bloomz-560m | MinED | 3.9 | 22.73±0.18 | 12.30±0.37 | 15.58±0.16 | 23.97±0.13 | 18.65±0.21 | 2.49% |
| | | ULD | 3.4 | 22.21±0.40 | 12.23±0.55 | 15.11±0.39 | 23.64±0.37 | 18.30±0.43 | 4.44% |
| | | GLD | 3.7 | 23.22±0.38 | **12.87±0.49** | **15.92±0.29** | **24.43±0.11** | **19.11±0.32** | - |

