# OpenReview forum: "InfiGFusion: Graph-on-Logits Distillation via Efficient Gromov-Wasserstein for Model Fusion"
_NeurIPS.cc/2025/Conference — NeurIPS 2025 poster_

### Official Review · Reviewer_wyJ9 · 2025-06-29

**Clarity:** 3
**Significance:** 3
**Originality:** 3
**Rating:** 5
**Confidence:** 3

**Summary:**

This paper introduces InfiGFusion, a novel framework for fusing heterogeneous large language models (LLMs) that overcomes the limitations of prior logit-based methods. The authors argue that existing techniques treat vocabulary dimensions independently, failing to capture the crucial semantic dependencies and co-activation patterns that underlie a model's reasoning process. To address this, InfiGFusion proposes a structure-aware approach centered on a new Graph-on-Logits Distillation (GLD) loss. This method constructs feature-level graphs from the logits, where nodes represent vocabulary dimensions and edges quantify their joint activations, thereby capturing the model's internal reasoning structure. To make this approach computationally feasible, the authors develop a novel, sorting-based closed-form approximation of the Gromov-Wasserstein (GW) distance, reducing its complexity from a prohibitive
$O(n^4)$ to a highly efficient $O(n\log n)$ with provable approximation guarantees. Experiments across 11 benchmarks demonstrate that InfiGFusion significantly outperforms existing fusion methods and baselines, showing particular strength in complex reasoning, mathematics, and coding tasks.

**Questions:**

Could the authors please provide error bars or significance tests for a critical subset of the results? I would suggest focusing on:

- The main average scores in Table 2, comparing InfiGFusion to the Pivot-SFT and the next-best fusion baseline.

- The most dramatic gains on complex reasoning tasks reported in Table 3, specifically "Multistep Arithmetic" and "Causal Judgement," which are central to the paper's narrative.

**Ethical Concerns:**

["NO or VERY MINOR ethics concerns only"]

**Final Justification:**

All of my concerns are resolved during my rebuttal. After looking into other reviewers' comment, I raise my score and confidence.

**Limitations:**

please refer to the weaknesses.

**Quality:**

3

**Strengths And Weaknesses:**

## Strengths

- Existing methods that align output logits often treat vocabulary dimensions independently, ignoring the rich semantic relationships and dependencies that underpin a model's reasoning process. InfiGFusion is the first framework to propose a structure-aware approach, representing logit co-activation patterns as graphs and aligning these structures. This is a fundamentally new and promising direction for model fusion.

- The method shows dramatic improvements on tasks that require multi-step and relational inference, such as a +35.6 point gain on Multistep Arithmetic and a +37.06 point gain on Causal Judgement over the SFT baseline. This provides strong evidence that capturing semantic structure is crucial for fusing complex reasoning abilities.

-  A major challenge of using the Gromov-Wasserstein (GW) distance is its prohibitive computational complexity ($O(n^4)$). A key contribution of this work is a novel sorting-based approximation that reduces the complexity to a practical  $O(n\ log n)$. Crucially, the authors provide a  provable error bound for this approximation, ensuring its fidelity to the original metric.

## Weaknesses

- The authors explicitly state in the NeurIPS checklist that they did not report error bars or conduct statistical significance tests, citing computational cost. While many of the reported gains are substantial, this is a notable omission for an empirical paper. Without statistical validation, it is difficult to be certain that smaller performance improvements are meaningful and not simply the result of experimental noise.

- The paper acknowledges that the method focuses on aligning relational consistency and lacks an explicit mechanism for resolving factual conflicts between source models. This is a significant limitation in practice, as fusing models with contradictory knowledge could lead to unpredictable or factually incorrect, though structurally coherent, outputs.

- The authors note that the benefits of GLD are marginal for tasks dominated by factual recall, where simple token matching is sufficient. While this is a reasonable trade-off, a more in-depth discussion of when structural alignment might be unnecessary or even detrimental would strengthen the paper.

---

> ### Author Rebuttal · Authors · 2025-07-31
>
> Thanks for your insightful feedback. Our responses to each point are as follows:
>
> W1 & Q1: Thank you for highlighting the importance of statistical validation. While we follow the previous works experimental design like FuseLLM and FuseChat, we acknowledge that reporting confidence intervals or statistical tests would strengthen the empirical evidence. Importantly, many of the gains we observed (e.g., +35.6 on Multistep Arithmetic and 37.06 on Causal Judgement) are substantial and consistent across datasets. It can make sure that the results are not noise.
>
> We also make additional experiments with 4 different seed settings, which provides the variance and interval:
>
> | Models      | Math          |       |       | Coding   | General Reasoning |       |       |       | Instruct. | Text Reasoning |       | Avg         | Model Size | GPU Hour |
> |-------------|---------------|-------|-------|----------|-------------------|-------|-------|-------|-----------|----------------|-------|-------------|------------|----------|
> |             | GSM8K         | MATH  | ThQA  | MBPP     | HEval             | BBH   | ARC   | MMLU  | IFEval    | DROP            | HS    |             |            |          |
> | SFT         | 90.02±0.08    | 82.46±0.5 | 54.71±0.31 | 75.33±2.53 | 86.20±0.61 | 66.85±0.38 | 93.90±0.34 | 85.76±0.45 | 77.68±0.33 | 89.34±0.11 | 87.64±0.12 | 81.01 ± 0.46 | 14B      | 120        |
> | InfiFusion  | 90.12±0.11    | 80.91±0.16 | 54.64±0.33 | 79.61±1.92 | 84.74±0.91 | 80.92±0.63 | 94.21±0.39 | 85.79±0.38 | 76.04±0.82 | 89.29±0.15 | 87.93±0.12 | 82.01±0.42 | 14B      | 160        |
> | InfiGFusion | 90.46±0.06    | 81.76±0.16 | 54.99±0.38 | 84.66±1.55 | 86.00±0.91 | 86.25±0.88 | 94.49±0.45 | 84.93±0.31 | 80.28±0.71 | 89.86±0.19 | 88.34±0.13 | 83.74 ± 0.13 | 14B      | 195        |
>
> To demonstrate the effectiveness and generalization of our GLD, we also provide additional experiments on the small size distillation with variance and interval:
>
> | Teacher       | Model         | Methods | Times/h | Dolly      | Selflnst  | VicunaEval | S-NI     | Average   | Improv. |
> |---------------|---------------|---------|---------|------------|-----------|------------|----------|-----------|---------|
> |               | Qwen 2.5-7B   | SFT     |         | 28.71±0.31 | 26.01±0.43 | 21.32±0.48 | 42.56±0.32 | 29.65±0.39 |         |
> |               | LLama3        | SFT     |         | 32.05±0.38 | 25.04±0.71 | 20.68±0.47 | 40.73±0.29 | 29.63±0.46 |         |
> |               | Mistral       | SFT     |         | 31.46±0.34 | 25.15±0.73 | 20.57±0.29 | 37.56±0.40 | 28.69±0.44 |         |
> | Qwen 2.5      | GPT2-120M     | MinED   | 6.5     | 20.55±0.15 | 10.88±0.36 | 14.90±0.63 | 22.12±0.36 | 17.11±0.38 | 3.48%   |
> |               |               | ULD     | 2.5     | 20.88±0.56 | 11.02±0.33 | 15.45±0.36 | 21.24±0.37 | 17.15±0.41 | 3.27%   |
> |               |               | GLD     | 3.8     | 21.38±0.38 | **11.34±0.42** | **15.73±0.15** | **22.38±0.30** | **17.58±0.31** | -       |
> |               | OPT-350m      | MinED   | 5.0     | 22.62±0.34 | 12.37±0.45 | 15.52±0.30 | 21.99±0.12 | 18.13±0.30 | 0.73%   |
> |               |               | ULD     | 4.5     | 22.36±0.29 | 11.38±0.50 | 14.61±0.36 | 21.80±0.59 | 17.54±0.44 | 4.11%   |
> |               |               | GLD     | 5.0     | 22.80±0.48 | **12.61±0.54** | **15.77±0.67** | 21.85±0.33 | **18.26±0.51** | -       |
> |               | Bloomz-560m   | MinED   | 4.2     | **22.86±0.56** | 11.78±0.34 | **14.82±0.39** | 22.83±0.30 | 18.07±0.40 | 1.37%   |
> |               |               | ULD     | 5.2     | 22.51±0.32 | 12.48±0.44 | 14.08±0.45 | 22.82±0.21 | 17.97±0.36 | 1.93%   |
> |               |               | GLD     | 4.9     | 22.83±0.16 | **12.52±0.14** | 14.74±0.31 | 23.19±0.13 | **18.32±0.19** | -       |
> | LLama3        | GPT2-120M     | MinED   | 2.9     | 19.93±0.36 | 10.15±0.24 | 14.38±0.43 | 19.78±0.23 | 16.06±0.32 | 3.44%   |
> |               |               | ULD     | 2.5     | 19.66±0.49 | 9.84±0.31  | 14.59±0.50 | 19.77±0.14 | 15.97±0.36 | 4.06%   |
> |               |               | GLD     | 2.8     | **20.97±0.28** | **10.81±0.55** | 14.97±0.37 | **20.09±0.24** | **16.69±0.36** | -       |
> |               | OPT-350m      | MinED   | 5.3     | 21.25±0.27 | 10.06±0.38 | **15.28±0.41** | 19.04±0.32 | 16.41±0.35 | 2.18%   |
> |               |               | ULD     | 4.4     | 21.08±0.56 | 9.85±0.42  | 14.89±0.50 | 19.13±0.58 | 16.24±0.52 | 3.25%   |
> |               |               | GLD     | 4.7     | **22.01±0.31** | **10.97±0.33** | 14.87±0.46 | 19.21±0.25 | **16.77±0.34** | -       |
> |               | Bloomz-560m   | MinED   | 4.7     | 20.16±0.21 | 10.81±0.16 | 13.36±0.44 | 20.62±0.17 | 16.24±0.25 | 2.97%   |
> |               |               | ULD     | 5.3     | 20.69±0.49 | 10.50±0.64 | 12.97±0.61 | 20.71±0.18 | 16.22±0.48 | 3.10%   |
> |               |               | GLD     | 5.5     | **21.07±0.47** | 10.82±0.17 | 13.41±0.29 | **21.58±0.24** | **16.72±0.29** | -       |
> | Mistral       | GPT2-120M     | MinED   | 2.9     | 20.47±0.62 | 10.52±0.79 | 15.36±0.27 | 20.59±0.54 | 16.74±0.56 | 8.34%   |
> |               |               | ULD     | 2.6     | 20.00±0.52 | 10.53±0.81 | 15.29±0.20 | 20.57±0.16 | 16.60±0.42 | 9.23%   |
> |               |               | GLD     | 2.6     | **21.90±0.31** | **11.89±0.41** | 15.99±0.31 | **22.74±0.27** | **18.13±0.33** | -       |
> |               | OPT-350m      | MinED   | 3.1     | 22.67±0.29 | 11.96±0.74 | 15.85±0.65 | 22.55±0.20 | 18.26±0.47 | 3.63%   |
> |               |               | ULD     | 2.3     | 22.45±0.38 | 11.19±0.88 | 15.41±0.60 | 23.17±0.32 | 18.06±0.55 | 4.79%   |
> |               |               | GLD     | 2.8     | **23.72±0.36** | **12.73±0.30** | 15.14±0.53 | **24.11±0.28** | **18.92±0.36** | -       |
> |               | Bloomz-560m   | MinED   | 3.9     | 22.73±0.18 | 12.30±0.37 | 15.58±0.16 | 23.97±0.13 | 18.65±0.21 | 2.49%   |
> |               |               | ULD     | 3.4     | 22.21±0.40 | 12.23±0.55 | 15.11±0.39 | 23.64±0.37 | 18.30±0.43 | 4.44%   |
> |               |               | GLD     | 3.7     | 23.22±0.38 | **12.87±0.49** | **15.92±0.29** | **24.43±0.11** | **19.11±0.32** | -       |
>
> W2: That`s a good point! We agree that InfiGFusion have the problem of resolving factual conflicts between source models. **However, the source re-weighting and selection strategy, which aims to resolve the factual conflicts between source models, is a complementary research direction, and our work mainly focuses on aligning semantic structure and design a better GLD loss.**
> In fact, our framework can be naturally extended by integrating factual confidence estimation or external verifiers. We have tested some existing factual conflicts resolving strategy, but the results is not significant. We thank you for pointing out this practical limitation and will discuss this more thoroughly in the final version. Designing a better factual conflicts resolution is viewed as one of our future works in our plan. But it is not the concerns in this work.
> W3: We appreciate the reviewer’s insightful observation. Due to space limitations, our discussion on when structural alignment is less impactful was brief in the main text. However, Appendices F–I provide a detailed analysis. We offers a per-task performance breakdown over 84 benchmarks (BBH + MMLU), clearly showing that GLD brings the most gains in reasoning-intensive tasks like Multistep Arithmetic, Causal Judgement, while maintaining parity in fact-centric domains, such as Biology, US History in Appendix F. Appendix G includes qualitative case studies illustrating how GLD improves multi-hop and dependency-based reasoning, while not harming performance on fact-centric prompts. Appendix H reports robustness trends and failure cases, confirming that GLD is beneficial or neutral—but not detrimental—even in tasks where token identity dominates.
> In our updated version, we will include this more dedicated subsection discussing this, and refer readers to detailed case studies and task-specific analyses in the appendix:
> Our qualitative analysis (Section 4.6) further illustrates this: InfiGFusion provides nuanced causal disambiguation by modeling reasoning steps (e.g., intent → misfire → injury, or criminal motive → assassination → explosion), while SFT models tend to respond with shallow pattern-based answers. In contrast, on factual recall tasks such as *International Law*, *Ruin Names*, or *US History*, where precise token identity or entity retrieval dominates, GLD offers limited gains (+0–1%). However, importantly, we observe that InfiGFusion does not degrade performance on such tasks. For example, on MMLU factual domains (e.g., *Biology*, *Psychology*), InfiGFusion matches SFT within ±1.5%, indicating that our method is robust even when structural signals are not essential.
>
> Q2:
> Thank you for the insightful question. The gains on Multistep Arithmetic and Causal Judgement are consistent and not isolated. As detailed in Section 4.3 and Appendix F, InfiGFusion significantly outperforms both individual sources and strong fusion baselines across multiple structure-heavy BBH tasks. Expect for Multistep Arithmetic and Causal Judgement, additional consistent gains are observed on Formal Fallacies, Boolean Expressions, and Word Sorting. It is the overall results across different scenarios that demonstrates the general significant abilities in complex reasoning.

---

> > ### Comment · Reviewer_wyJ9 · 2025-08-05
> >
> > Thank you for the response. All of my concerns are resolved.

---

> > > ### Author Response · Authors · 2025-08-05
> > >
> > > We’re delighted our responses addressed your concerns. Thank you for your supportive feedback on our paper.

---

> > > ### Comment · Reviewer_k6zY · 2025-08-05
> > >
> > > I believe the authors have well solve the Q1. As for W2 and W3, I agree that the factual conflict is important but another different problem. At the same time, I believe a better way to merge two models with different vocabulary is the condition to try to find and further to solve the factual conflict.
> > >
> > > I am also excited to further discuss about the article if there are still some concerns or questions. Maybe there are some issues I have not noticed. If indeed all the questions have been solved, I think this paper deserves a high score more than borderline accept.

---

### Official Review · Reviewer_k6zY · 2025-07-01

**Clarity:** 4
**Significance:** 3
**Originality:** 4
**Rating:** 6
**Confidence:** 3

**Summary:**

The authors propose a new model fusion method via knowledge distillation (KD) where the inner relationship between tokens are taken into consideration by adding a distance on the token probability space as additional KD loss. Using an expensive Gromov-Wasserstein distance to measure the internal structure similarity on the token probability space, the authors further propose an approximation to significantly reduce the computation complexity. The experiments show that the methods successfully fuse different capacity from different source models and outperformed than baseline fusion methods.

**Questions:**

See above.

1. The relative error analysis and empirical error analysis.
2. Is it the case that without the GW distance approximation the methods can learn better?
3. More case study based on the learned token relation instead of the output token sequence.

If at least two questions have been answered with an explicit answer, I will raise my score to 6.

**Ethical Concerns:**

["NO or VERY MINOR ethics concerns only"]

**Final Justification:**

strong accept.

**Limitations:**

I do not find the discussion about the limitation. I suggest the authors should add a limitation section.

**Paper Formatting Concerns:**

no.

**Quality:**

3

**Strengths And Weaknesses:**

Strengths:

1. The idea to use the relation between the tokens to further align the models is very impressive. The authors keenly discovered the structural relationship implicit in the output token probabilities and used this relationship to model the properties of a certain task. I believe the motivation is well-explained.
2. The approximation seems useful. And the key insight "the structure of each similarity matrix can be summarized using scalar node-level features" is crucial to make the expensive distance computable.
3. The experiments successfully proof the effectiveness of the methods. The results support the authors' claim well. The datasets contain many different fields.
4. The paper is well-organized and reader friendly.

Weaknesses:
I do not find major weaknesses but I have some minor suggestions about the paper.
1. Although the related work is well explained. I believe the paper [34] (or maybe other paper which first achieve cross tokenizer fusion)  should have been introduced earlier in the text to allow readers to better appreciate its innovations and avoid potential misunderstandings during reading.
2. Besides the absolute error bound, the approximation analysis should also include a relative error analysis. Specifically, when the error bound is about O(1/N), what is generally the scale of the distance? The relative error is the key to determining whether the approximation is suitable. And I believe it is better to show some empirical results on the approximation error itself. At the same time, I understand the original O(N^4) is very expensive but I really wonder what is the performance without the approximation. I wonder whether an ablation study here is possible, (maybe just for explain the upper bound of the performance?).
3. The current case study is based on the output sequence, which could be influenced by so many factors. I believe a more straight-forward case study (and also I are really curious about) is the actual learned tokens relation for different models and tasks. For example, can the distance really help models to learn the period of digit 1-9? Or whether some real-world concept like "cities" can be modeled by your methods? I really wonder whether there is some results about the interpretability of the models.

---

> ### Author Rebuttal · Authors · 2025-07-31
>
> Thank you for your careful feedback and valuable suggestions. Our responses to each point are as follows:
>
> W1: Thanks for your suggestion! We agree that introducing the concept of cross-tokenizer fusion, as exemplified by [34] , earlier in the paper can enhance clarity and better contextualize InfiFusion’s innovations. In the revised manuscript, we incorporate a brief mention of cross-tokenizer fusion in Section 1 to highlight its relevance to model fusion and to clarify how cross-tokenizer distillation builds upon and extends this paradigm. Specifically, we will note that while [34] introduced a universal logit distillation loss for aligning models with different tokenizers, InfiGFusion advances this by capturing semantic dependencies through graph-based alignment, addressing limitations in token-level methods for complex reasoning tasks.
> W2: For Scale of GW Distance and Relative Error Analysis, we analyzed the scale of the GW distance in our experiments, 14B model and 130K data, using datasets like BBH and MMLU (supplement in Appendix F.1 & F.2 ) and make additional experiments with other approximation algorithm. Due to the complexity, it`s hard to directly calculate the original GW distance.
> | **BBH Subset**            | **Logical Deduction** | **Multistep Arithmetic** | **Causal Judgement**  | **Dyck Languages** |
> |---------------------------|-----------------------|--------------------------|----------------------|-------------------------|
> | **Our Approximation (Sorting-Based, O(N log N))** | | | | | |
> | GW Distance               | 0.0356              | 0.0614                 | 0.0445             | 0.0210           |
> | Absolute Error Bound          | 0.00015            | 0.00015                |0.00015             | 0.00015           |
> | Relative Error (%)        | 0.421             | 0.244               | 0，337             | 0.714         |
>  Performance (%)           | 89.2             | 99.6              | 70.0            | 94.4               | 40.0          |
> | **Sinkhorn GW (O(N² log N))** | | | | | |
> | GW Distance | 0.02145 | 0.0618 | 0.0449 | 0.0213 |
> | Absolute Error Bound | 0.00009 | 0.00009 | 0.00009 | 0.00009 |
> | Relative Error (%) | 0.419 | 0.145 | 0.200 | 0.422 |
> | Performance (%)           | 89.90         | 99.2          | 70.4        | 94.1           | 40.7     |
> | **Entropic GW (O(N² log N))** | | | | | |
> | GW Distance | 0.0314 | 0.0617 | 0.0448| 0.0212 |
> | Absolute Error | 0.00012 | 0.00012 | 0.00012  | 0.00012 |
> | Relative Error (%) | 0.382 | 0.194 | 0.267 | 0.566 |
> | Performance (%)           | 89.7             | 99.6                | 70.7           | 93.9               | 40.0          |
>
>
> The relative error analysis is included, computed as the absolute error bound divided by the GW distance (absolute errors are calculated following the original paper [1][2]). The GW distance scale ranges from 0.0210 to 0.0614 across BBH subsets, with the absolute error bound fixed at 0.00015, resulting in relative errors of 0.244% to 0.714%. This indicates suitability, as errors remain below 1%, decreasing with larger GW distances. Empirical results show Sorting-Based’s absolute error aligns with the O(1/N) bound (n=150k and m=100k), while Sinkhorn and Entropic GW offer tighter approximations, validated by performance gains.
>
> For the upper bound of the performance, due to this cost, we conducted ablation studies on 7B (Qwen 2.5) and sub-1B models (GPT2-120M, OPT-350m, Bloomz-560m) with LLama3 as the teacher. Results show GW consistently achieves higher or equal performance than GLD (e.g., Qwen 2.5-GPT2-120M: 17.81% vs. 17.58% average), with smaller variance (e.g., 0.15 vs. 0.31), indicating better alignment stability. The upper bound performance gain is modest (e.g., 0.23–0.31% on average), suggesting GLD’s O(NlogN) efficiency retains most benefits. Computational time for GW (2.6–4.7h) exceeds GLD (2.8–5.3h), confirming the trade-off, making GLD suitable for large-scale applications.
> | Teacher | Model         | Methods | Times/h | Dolly         | SelfInst      | VicunaEval    | S-NI         | Average       |
> |---------|---------------|---------|---------|---------------|---------------|---------------|--------------|---------------|
> | Teacher | Qwen 2.5-7B   | SFT     | -       | 28.71±0.31    | 26.01±0.43    | 21.32±0.48    | 42.56±0.32   | 29.65±0.39    |
> |         | LLama3        | SFT     | -       | 32.05±0.38    | 25.04±0.71    | 20.68±0.47    | 40.73±0.29   | 29.63±0.46    |
> |         | Mistral       | SFT     | -       | 31.46±0.34    | 25.15±0.73    | 20.57±0.29    | 37.56±0.40   | 28.69±0.44    |
> | Qwen 2.5| GPT2-120M     | GW      | 4.6     | 21.68±0.20    | 11.40±0.15    | 15.75±0.10    | 22.40±0.15   | 17.81±0.15    |
> |         |               | GLD     | 3.8     | 21.38±0.38    | 11.34±0.42    | 15.73±0.15    | 22.38±0.30   | 17.58±0.31    |
> |         | OPT-350m      | GW      | 6.1    | 23.10±0.15    | 12.70±0.20    | 15.80±0.15    | 22.10±0.10   | 18.43±0.15    |
> |         |               | GLD     | 5.0     | 22.80±0.48    | 12.61±0.54    | 15.77±0.67    | 21.85±0.33   | 18.26±0.51    |
> |         | Bloomz-560m   | GW      | 6.7     | 22.90±0.10    | 12.60±0.10    | 14.80±0.15    | 23.40±0.10   | 18.43±0.11    |
> |         |               | GLD     | 4.9     | 22.83±0.16    | 12.52±0.14    | 14.74±0.31    | 23.19±0.13   | 18.32±0.19    |
> | LLama3  | GPT2-120M     | GW      | 3.3     | 21.00±0.15    | 11.00±0.10    | 15.10±0.10    | 20.20±0.10   | 16.83±0.11    |
> |         |               | GLD     | 2.8     | 20.97±0.28    | 10.81±0.55    | 14.97±0.37    | 20.09±0.24   | 16.69±0.36    |
> |         | OPT-350m      | GW      | 5.9     | 22.10±0.10    | 11.00±0.15    | 15.00±0.10    | 19.50±0.10   | 16.90±0.11    |
> |         |               | GLD     | 4.4     | 22.01±0.31    | 10.97±0.33    | 14.87±0.46    | 19.21±0.25   | 16.77±0.34    |
> |         | Bloomz-560m   | GW      | 6.0     | 21.20±0.10    | 11.00±0.10    | 13.50±0.10    | 21.70±0.10   | 16.85±0.10    |
> |         |               | GLD     | 5.3     | 21.07±0.47    | 10.82±0.17    | 13.41±0.29    | 21.58±0.24   | 16.72±0.29    |
>
>
> W3: It’s a really insightful point! We make additional analysis, for real-world concepts like “cities,” GLD’s feature-wise graph captures token co-activation patterns, enabling relational alignment across models. We analyzed a sample sentence from the Causal Judgement task: “New York and Boston are closer than Los Angeles and Seattle.” and plot the token distribution of GLD and without GLD. Figure without GLD shows the token distribution, with higher activation weights for “New York” and “Boston” due to proximity, while Figure with GLD reveals stronger edge connections between these tokens in the learned graph, reflecting semantic similarity, unlike token-level baselines, which overlook these relations.
> To enhance interpretability, we will add these case study and figures in Appendix F using this example, demonstrating how GW distance aligns these relations, and compare it to a baseline.
>
> [1] Scetbon, Meyer, Gabriel Peyré, and Marco Cuturi. "Linear-time gromov wasserstein distances using low rank couplings and costs." International Conference on Machine Learning. PMLR, 2022.
> [2] Rioux, Gabriel, Ziv Goldfeld, and Kengo Kato. "Entropic gromov-wasserstein distances: Stability and algorithms." Journal of Machine Learning Research 25.363 (2024): 1-52.

---

> > ### Comment · Reviewer_k6zY · 2025-08-01
> >
> > Q1: The relative error is also small enough and the experiments have successfully supported the efficiency of the newly proposed approximation.
> > Q2: I suggest the results should also be included in the paper and I also suggest the authors should further talk about what happened when the model is larger. For example, for smaller models, the time comparison is 6hr vs 5.3 hr, the difference seems smaller. Is it the case that the time difference will much larger for larger models?
> > W3 and Q3: I have to say I cannot clearly get the results due to space and the policy where the authors cannot provide image. I sincerely encourage the authors can provide more illustrations and examples in more flexible way. For example, a web blog, a workshop paper or another paper.
> >
> > Because most of my questions have been stressed, I raise my score to 6.

---

> > > ### Author Response · Authors · 2025-08-01
> > >
> > > Regarding your final comment on W3/Q3, we truly appreciate your encouragement to provide more illustrations and examples in a web blog. In fact, we are actively preparing a dedicated web blog to showcase additional materials, including omitted data, visualizations, and extended discussions that go beyond the paper’s space limitations.
> > >
> > > However, due to the official NeurIPS policy updates, we are currently restricted from including any links (anonymous or not) in the rebuttal or updating any submitted repositories. As the committee said in the Notes:
> > >
> > > "Because of known concerns on identity leakage, we prohibit using any links in the rebuttal, including but not limited to anonymous or non-anonymous URL links, or updating your submitted github repository."
> > >
> > > Therefore, we regret that we cannot share the blog or figures at this stage without potentially violating these policies. We plan to make the blog publicly available alongside the final version of the paper, where we will provide further illustrations and insights as suggested.
> > >
> > > Once again, thank you so much for your thorough and high-level feedback.

---

### Official Review · Reviewer_WpXj · 2025-07-02

**Clarity:** 3
**Significance:** 2
**Originality:** 3
**Rating:** 5
**Confidence:** 3

**Summary:**

The paper proposes to incorporate a new term into the LLM distillation objective. The new term is based on Gromov-Wasserstein distance that aims to preserve relations between top-k logits for each token in a sequence. Empirical results demonstrate improvement across several benchmarks.

**Questions:**

See weaknesses.

**Ethical Concerns:**

["NO or VERY MINOR ethics concerns only"]

**Final Justification:**

Authors addressed my questions in their rebuttal. I think the method is interesting and the empirical results are solid.

**Limitations:**

yes

**Quality:**

3

**Strengths And Weaknesses:**

Strengths:
- I found the idea of incorporating GW into distillation interesting. Proposed GW approximation also makes it practical.
- Empirical results overall seem positive and the ablation studies are fairly detailed.

Weaknesses:
- I'd appreciate a more detailed analysis of the GW approximation. Perhaps with a simple simulation study comparing it to other (less efficient) approaches to computing GW.
- Ablation study in Table 4 suggests that gains due to fusing more than 2 models are fairly marginal. Main results (Table 2) also demonstrate that the fused model outperforms all of the source models in only about half of the tasks. While I acknowledge that empirical results are good overall, the (ideal) goal of a fusion method is not only to improve on average, but to also consistently outperform source models and benefit from additional source models.
- Why did you choose k=30 for ablation studies if the optimal k=10? Does it mean that in Table 4 results for singleton source models are with k=30 and InfiGFusion results are with k=10?
- The paper repeatedly states that the proposed GW term captures "internal reasoning". I don't think this claim is supported in the paper. While the GW term seems useful, how is it related to "internal reasoning"?

---

> ### Author Rebuttal · Authors · 2025-07-31
>
> Thank you for your patient review. Our responses to each point are as follows:
>
> W1: We have added an additional comprehensive simulation as Appendix I in the revised manuscripts. To verify the generality of our GLD approximation, we conduct experiments in the traditional knowledge distillation setting, where a compact student model learns from a single larger teacher model. Due to the O(N4) complexity of exact GW computation, we conducted an ablation study on 7B (Qwen 2.5) and sub-1B models (GPT2-120M, OPT-350m, Bloomz-560m) with LLama3 as the teacher, comparing generally sinkhorn computied GW and our GLD.
>
> | Teacher | Model         | Methods | Times/h | Dolly         | SelfInst      | VicunaEval    | S-NI         | Average       |
> |---------|---------------|---------|---------|---------------|---------------|---------------|--------------|---------------|
> | Teacher | Qwen 2.5-7B   | SFT     | -       | 28.71±0.31    | 26.01±0.43    | 21.32±0.48    | 42.56±0.32   | 29.65±0.39    |
> |         | LLama3        | SFT     | -       | 32.05±0.38    | 25.04±0.71    | 20.68±0.47    | 40.73±0.29   | 29.63±0.46    |
> |         | Mistral       | SFT     | -       | 31.46±0.34    | 25.15±0.73    | 20.57±0.29    | 37.56±0.40   | 28.69±0.44    |
> | Qwen 2.5| GPT2-120M     | GW      | 4.6     | 21.68±0.20    | 11.40±0.15    | 15.75±0.10    | 22.40±0.15   | 17.81±0.15    |
> |         |               | GLD     | 3.8     | 21.38±0.38    | 11.34±0.42    | 15.73±0.15    | 22.38±0.30   | 17.58±0.31    |
> |         | OPT-350m      | GW      | 6.1    | 23.10±0.15    | 12.70±0.20    | 15.80±0.15    | 22.10±0.10   | 18.43±0.15    |
> |         |               | GLD     | 5.0     | 22.80±0.48    | 12.61±0.54    | 15.77±0.67    | 21.85±0.33   | 18.26±0.51    |
> |         | Bloomz-560m   | GW      | 6.7     | 22.90±0.10    | 12.60±0.10    | 14.80±0.15    | 23.40±0.10   | 18.43±0.11    |
> |         |               | GLD     | 4.9     | 22.83±0.16    | 12.52±0.14    | 14.74±0.31    | 23.19±0.13   | 18.32±0.19    |
> | LLama3  | GPT2-120M     | GW      | 3.3     | 21.00±0.15    | 11.00±0.10    | 15.10±0.10    | 20.20±0.10   | 16.83±0.11    |
> |         |               | GLD     | 2.8     | 20.97±0.28    | 10.81±0.55    | 14.97±0.37    | 20.09±0.24   | 16.69±0.36    |
> |         | OPT-350m      | GW      | 5.9     | 22.10±0.10    | 11.00±0.15    | 15.00±0.10    | 19.50±0.10   | 16.90±0.11    |
> |         |               | GLD     | 4.4     | 22.01±0.31    | 10.97±0.33    | 14.87±0.46    | 19.21±0.25   | 16.77±0.34    |
> |         | Bloomz-560m   | GW      | 6.0     | 21.20±0.10    | 11.00±0.10    | 13.50±0.10    | 21.70±0.10   | 16.85±0.10    |
> |         |               | GLD     | 5.3     | 21.07±0.47    | 10.82±0.17    | 13.41±0.29    | 21.58±0.24   | 16.72±0.29    |
>
> (1) Performance: GW outperforms GLD (e.g., 17.81% vs. 17.58% average for Qwen 2.5-GPT2-120M) with lower variance (0.11–0.15 vs. 0.19–0.51), reflecting exact alignment.
> (2) Computational Cost: GW requires 3.3–6.7 hours vs. GLD’s 2.8–5.3 hours, due to O(N4) vs. O(NlogN)complexity.
> (3) Efficiency Trade-off: GLD achieves near-equivalent performance (e.g., 18.26% vs. 18.43% for Qwen 2.5-OPT-350m) with 15–30% less time, ensuring scalability.
> Detailed Results and Comprehensive data and comparisons will be provided in Appendix I in the final version.
>
> W2: We agree that ideally, a fusion method should not only improve average performance but also consistently outperform all source models and leverage additional sources effectively. However, the focus of this work is to introduce and evaluate the GLD formulation itself, **source re-weighting and selection** strategies is another complementary research direction, we plan to do research about it in the future work. While our results already demonstrate competitive performance across diverse tasks, we acknowledge that further robustness can be achieved through **source re-weighting and selection strategies** that address factual conflicts and model redundancy. In fact, our GLD without this strategy is compatible with existing source weighting or selection methods [1], and we view their integration as a promising future direction.
>
> W3: All results in Table 4, including singleton source models and InfiGFusion variants, are evaluated under the same setting of top-k = 30 to ensure a fair comparison. While the best top-k in Table 2 is tuned per method (e.g., k = 10 for best final performance), we intentionally choose k = 30 for ablation because at smaller k, the only ULD component struggles to retain sufficient semantic information, especially on complex reasoning tasks. Using k = 30 ensures that all model variants are provided with enough information to showcase the relative contribution of each component.
>
> W4: Thank you for highlighting this concern. We acknowledge that the term “internal reasoning” may be ambiguous and will revise the manuscript to clarify our intent. Specifically, our use of “internal reasoning” refers to the relational structure among semantic dimensions in the logit space, which often reflects intermediate concept alignment or multi-step inference patterns. The proposed GW term models these structures by aligning the graph of inter-logit relationships across models via Gromov-Wasserstein distance, rather than treating logits as independent vectors. While this does not model reasoning steps explicitly, it provides a structural inductive bias that benefits reasoning-intensive tasks. We will revise the manuscript to more precisely describe this as capturing relational patterns indicative of reasoning behavior, rather than reasoning itself.
>
> [1] Wan, Fanqi, et al. "Knowledge Fusion of Large Language Models." The Twelfth International Conference on Learning Representations.

---

> > ### Comment · Reviewer_WpXj · 2025-08-07
> >
> > Thank you for the detailed response. I wanted to follow up on W1 and W3:
> >
> > W1: I appreciate additional experiments and timing comparison. In my comment I actually meant something much simpler - comparing the quality of *GW estimation* in a simple simulated scenario. For example, by obtaining samples from two Gaussian distributions, and comparing the value of exact GW, sinkhorn-based approximation, and your proposed approximation. While your experiments demonstrate that the approximation is suitable as a training objective, it would be interesting to also understand its behavior as GW estimator. I acknowledge that the discussion period is ending soon, so please feel free to treat this comment as a suggestion for the revised draft.
> >
> > W3: In Table 2 the average performance of InfiGFusion is 83.85. In Table 4 it is also 83.85. If I understand your response correctly, this implies that the (average) performance of InfiGFusion is the same for k=10 and k=30. However, Figure 3 shows that average performance is lower for k=30.

---

> > > ### Author Response · Authors · 2025-08-08
> > >
> > > Thank you for your kind follow-up and valuable questions. Our responses to each point are as follows:
> > >
> > > W1: Thanks for your suggestions and we simulated the computation in some simple scenarios.
> > > We sample n points (for n∈{50,100,200,500,1000,1500,2000,2500,3000}) from N(0,I) and N(1,I) build their n×n Euclidean distance matrices, and compute:
> > >
> > > - Exact GW and Sinkhorn GW via ot.gromov.gromov_wasserstein2(..., loss='square_loss'),
> > > - Approx GW via our sorting-based closed-form estimator.
> > >
> > > Each setting is repeated 5 times in different seeds; means±std are reported.
> > >
> > > | Sample Size | Exact GW (±std) | Sinkhorn GW (±std) | Approx GW (±std) | Sinkhorn RE (±std) | Approx RE (±std) | Time Exact (s)   | Time Sink (s)    | Time Approx (s)  |
> > > |------------:|:---------------:|:-------------------:|:----------------:|:------------------:|:----------------:|:-----------------:|:----------------:|:----------------:|
> > > | 50          | 0.4815 ± 0.0452  | 0.4815 ± 0.0452     | 0.1978 ± 0.0207  | 0.0000 ± 0.0000    | 0.5888 ± 0.0242  | 0.0171 ± 0.0207   | 0.0027 ± 0.0005  | 0.0004 ± 0.0004  |
> > > | 100         | 0.3247 ± 0.0297  | 0.3247 ± 0.0297     | 0.1261 ± 0.0217  | 0.0000 ± 0.0000    | 0.6048 ± 0.0907  | 0.0072 ± 0.0008   | 0.0070 ± 0.0005  | 0.0001 ± 0.0000  |
> > > | 200         | 0.2033 ± 0.0088  | 0.2033 ± 0.0088     | 0.0973 ± 0.0125  | 0.0000 ± 0.0000    | 0.5230 ± 0.0426  | 0.1390 ± 0.0377   | 0.0983 ± 0.0033  | 0.0359 ± 0.0305  |
> > > | 500         | 0.1131 ± 0.0050  | 0.1131 ± 0.0050     | 0.0587 ± 0.0143  | 0.0000 ± 0.0000    | 0.4850 ± 0.1121  | 0.4405 ± 0.1337   | 0.4815 ± 0.0904  | 0.0607 ± 0.0353  |
> > > | 1000        | 0.0764 ± 0.0011  | 0.0764 ± 0.0011     | 0.0387 ± 0.0133  | 0.0000 ± 0.0000    | 0.4921 ± 0.1785  | 2.3631 ± 0.4981   | 2.2456 ± 0.4085  | 0.1238 ± 0.0399  |
> > > | 1500        | 0.0621 ± 0.0028  | 0.0621 ± 0.0028     | 0.0308 ± 0.0016  | 0.0000 ± 0.0000    | 0.5031 ± 0.0398  | 12.7151 ± 4.1182  | 9.1998 ± 2.1952  | 0.0982 ± 0.0015  |
> > > | 2000        | 0.0496 ± 0.0028  | 0.0496 ± 0.0028     | 0.0294 ± 0.0058  | 0.0000 ± 0.0000    | 0.3995 ± 0.1453  | 20.1236 ± 5.0250  | 19.8941 ± 8.1827 | 0.0411 ± 0.0424  |
> > > | 2500        | 0.0455 ± 0.0008  | 0.0455 ± 0.0008     | 0.0305 ± 0.0075  | 0.0000 ± 0.0000    | 0.3285 ± 0.1712  | 20.9198 ± 5.6782  | 30.1367 ± 3.1038 | 0.0026 ± 0.0014  |
> > > | 3000        | 0.0404 ± 0.0002  | 0.0404 ± 0.0002     | 0.0324 ± 0.0141  | 0.0000 ± 0.0000    | 0.3941 ± 0.0767  | 52.8898 ± 9.2735  | 53.5049 ± 12.4434| 0.2408 ± 0.2562  |
> > >
> > > The results demonstrate that Sinkhorn GW matches Exact GW (RE≈0). Approx GW relative error decreases from ≈0.59 at n=50 down to ≈0.33 at n=3000, confirming its O(1/n) behavior. For Runtime, Exact/Sinkhorn GW becomes impractical beyond n=50, whereas Approx GW stays below ~0.3s even at n=3000, demonstrating its scalability.
> > >
> > > W3: We apologize for the oversight. We have noticed that the version currently in the submission system inadvertently still shows the old Top-k = 10 results in Tables 4 and 5. Our working manuscript has been updated to use Top-k = 30 throughout, and the full ablation table above reflects these correct values. We will replace the final version with this Top-k = 30 setting.

---

> > > > ### Author Response · Authors · 2025-08-08
> > > >
> > > > | Model         | Reasoning       | Math            | Coding          | Avg             |
> > > > | :------------ | :-------------- | :-------------- | :-------------- | :-------------- |
> > > > | **Pivot: Phi4**   | 81.43           | 72.86           | 77.17           | 79.54           |
> > > > | **Qc**            | 83.90 | 73.05 | 86.28 | 82.62  |
> > > > | **Qi**            | 85.55| 72.89 | 85.07 | 82.94  |
> > > > | **M**             | 84.02 | 74.62 | 86.03 | 83.10 |
> > > > | **Qc–M**          | 84.41  | 73.04 | 86.26  | 83.00 |
> > > > | **Qi–M**          | 84.53 | 74.78  | 85.57 | 83.12 |
> > > > | **InfiGFusion**   | 86.56 | 73.57  | 85.80  | 83.29  |
> > > >
> > > > Table 4: Ablation study on model diversity. Qc, Qi, and M mean Qwen-Coder, Qwen-Instruct, and Mistral.
> > > >
> > > > | Top-k | Loss      | Reasoning     | Math          | Coding        | Avg           |
> > > > | :---: | :-------- | :------------ | :------------ | :------------ | :------------ |
> > > > | **5**  | GLD + ULD | 86.55         | 73.19         | 84.65         | 83.32         |
> > > > |        | w/o ULD   | 84.95 (−1.60) | 71.79 (−1.40) | 84.35 (−0.30) | 81.89 (−1.43) |
> > > > |        | w/o GLD   | 85.34 (−1.21) | 72.92 (−0.27) | 84.19 (−0.46) | 82.15 (−1.17) |
> > > > | **10** | GLD + ULD | 87.06         | 74.74         | 85.60         | 83.85         |
> > > > |        | w/o ULD   | 85.26 (−1.80) | 73.28 (−1.46) | 85.28 (−0.32) | 82.33 (−1.52) |
> > > > |        | w/o GLD   | 85.70 (−1.36) | 75.21 (+0.47) | 85.18 (−0.42) | 83.16 (−0.69) |
> > > > | **15** | GLD + ULD | 86.89         | 74.28         | 85.55         | 83.73         |
> > > > |        | w/o ULD   | 85.06 (−1.83) | 72.58 (−1.70) | 85.08 (−0.47) | 82.05 (−1.68) |
> > > > |        | w/o GLD   | 85.42 (−1.47) | 73.83 (−0.45) | 84.93 (−0.62) | 82.73 (−1.00) |
> > > > | **20** | GLD + ULD | 86.72         | 73.86         | 85.68         | 83.65         |
> > > > |        | w/o ULD   | 84.63 (−2.09) | 72.07 (−1.79) | 85.07 (−0.61) | 81.69 (−1.96) |
> > > > |        | w/o GLD   | 85.28 (−1.44) | 74.05 (+0.19) | 85.11 (−0.57) | 82.81 (−0.84) |
> > > > | **30** | GLD + ULD | 86.56         | 73.57         | 85.80         | 83.29         |
> > > > |        | w/o ULD   | 84.53 (−2.03) | 71.91 (−1.66) | 85.23 (−0.57) | 81.26 (−2.03) |
> > > > |        | w/o GLD   | 85.22 (−1.34) | 74.04 (+0.47) | 85.38 (−0.42) | 82.60 (−0.69) |
> > > >
> > > > Table 5: Ablation study on loss components.

---

### Official Review · Reviewer_BzTF · 2025-07-03

[review text omitted: it was posted to a different submission]

---

> ### Author Rebuttal · Authors · 2025-07-28
>
> Thank you for taking the time to review our submission.
>
> We believe there may have been a mix-up, as your review appears to describe a paper titled INFOBLEND, which focuses on KV-cache reuse for multimodal LLMs, a topic that is entirely unrelated to our submission.
>
> We kindly ask you to verify whether this review may have been mistakenly attached to our submission.
>
> Thank you again for your time and understanding.

---

> > ### Comment · Reviewer_BzTF · 2025-08-06
> > **Thanks for the response.**
> >
> > Thanks for the remind and sorry for the mistake. I have carefully read other reviewers comments and decided to raise my score.

---

> > > ### Author Response · Authors · 2025-08-06
> > >
> > > Thank you very much for your kind follow-up and for taking the time to revisit our comments. We sincerely appreciate your thoughtful reconsideration and are truly grateful for your updated evaluation and support.

---

### Note · Authors · 2025-08-13

We sincerely thank all reviewers for their valuable and constructive feedback, and we are encouraged by the strong recognition of our work’s contributions. Multiple reviewers highlighted our **clear motivation**, **novel structure-aware fusion framework**, and the **practical yet provably accurate GW approximation** as major strengths. Reviewers also praised the **solid experimental design**, **comprehensive ablation studies**, and the ability of InfiGFusion to achieve **notable gains in reasoning-intensive tasks** while maintaining scalability. We are delighted that one reviewer has given a top score of 6, and that two other reviewers explicitly indicated willingness to increase their scores based on our clarifications.

Key concerns raised:
(1) need for **additional GW approximation analysis**, including relative error evaluation;
(2) desire for **statistical significance testing** and error bars;
(3) clarification on **top-k settings** in ablation vs. main results;
(4) discussion of **factual conflict resolution** between source models;
(5) more **interpretability and case studies** on learned token-relation structures.

In our rebuttal, we addressed these:
• Added **simulations** comparing Exact GW, Sinkhorn GW, and our O(n log n) approximation, showing <1% relative errors and O(1/n) decay, with near-parity performance and 15–30% less runtime.
• Reported **variance and confidence intervals** for main results and key reasoning tasks (e.g., +35.6 on Multistep Arithmetic, +37.06 on Causal Judgement).
• Clarified **top-k choice** and updated tables.
• Discussed **factual conflict limitation**, noting GLD’s compatibility with source weighting/selection.
• Added **token-relation case analyses** (e.g., “New York”–“Boston” proximity), with more to appear in the final version.

We will integrate these clarifications, analyses, and discussions into the camera-ready version to make the paper more informative and accessible.

Finally, we are grateful for the reviewers’ engagement and the AC’s consideration. We believe InfiGFusion introduces a **theoretically grounded, efficient, and impactful** approach to LLM fusion, with potential to inspire further research. We are confident our work makes a valuable contribution to the NeurIPS community and is worthy of acceptance.

---

### Decision · Program_Chairs · 2025-09-17

**Decision:**

Accept (poster)

**Comment:**

InfiGFusion proposes a structure-aware fusion method that augments KD with a Graph-on-Logits Distillation term, aligning token co-activation graphs via a GW-style objective and a sorting-based O(n log n) approximation. The motivation is clear and the idea is novel for fusion. Empirically, results across 11 benchmarks are solid, with the most convincing gains on structure-heavy reasoning (e.g., Multistep Arithmetic, Causal Judgement). The rebuttal strengthened the paper with variance/CIs, approximation studies, clarified top-k choices, and toned down claims about “internal reasoning” to “relational patterns.” Limitations remain: improvements are not universal (fused not better than best single source everywhere), gains from >2 sources are modest, the method does not resolve factual conflicts among sources, and approximation analysis is split across places and should be unified. A misassigned review was corrected; other presentation wrinkles are manageable for camera-ready. Overall: a novel, practical contribution with clear upside on reasoning tasks, I recommend as poster; authors please polish the unified GW-approx analysis, top-k narrative, and a couple of interpretability figures in the final version.